# Host genome and bacterial taxa shape the Arabidopsis seed microbiome

Sabiha Parween[1,4], Naheed Tabassum [ID][1,4], Kirti Shekhawat [ID][1], Bruno Gnannt[2], Waad Alzayed [ID][1], Rewaa Jalal[3] & Heribert Hirt [ID][1✉]

## Abstract

Plant-microbiome interactions are crucial in shaping plant growth, stress resilience, and disease resistance. Among these, the seed microbiome plays a pivotal role in early plant development and ecological adaptation. However, little is known about the factors that determine the abundance and functions of the seed microbiome, as well as the role of the host genome in shaping the microbial diversity across different ecotypes. In this study, we investigated the diversity of the Arabidopsis seed microbiomes that originate from multiple geographical locations. High-throughput sequencing identified key bacterial taxa that govern Arabidopsis seed microbiota diversity. Distinct compositions of bacterial taxa were identified in Arabidopsis accessions sharing geographical location and similar soil features. Genome-wide association studies (GWAS) revealed that both the abundance of key taxa and common functional traits are associated with specific host genetic loci such as the RNA-binding protein RPB47B, mutants of which showed altered physiological properties related to soil properties and microbial diversity. Overall, our study establishes that geographical, soil and genetic host factors shape the Arabidopsis seed microbiome.

**Keywords** Arabidopsis; Seed Microbiome; Stress Fitness; GWAS
**Subject Categories** Microbiology, Virology & Host Pathogen Interaction; Plant Biology

## Introduction

Plants are holobionts, thriving through intricate interactions with their associated microbial communities (Saarenpää et al, 2024). Over the past few decades, the scientific community has increasingly focused on understanding the plant-associated microbiome both aboveground (leaves, flowers, fruits, seeds) and belowground (roots, rhizoplane, rhizosphere) environments (Abdelfattah et al, 2021; Sohrabi et al, 2023; Trivedi et al, 2020; Wei et al, 2021; Xiong et al, 2021; Zeng et al, 2023) A critical area of interest is how in-situ geo-climatic conditions, soil properties, and the host plant species shape the composition of these native microbial communities, with a particular focus on understanding the degree to which the host influences microbiome composition across populations of a single species. The complexity and dynamics of the Arabidopsis microbiome, both above and below ground, have been studied extensively in various contexts. For example, root-associated beneficial bacterial communities in *Arabidopsis thaliana* are essential for maintaining a balanced microbial ecosystem conducive to plant health (Durán et al, 2018). Specific microbial taxa, such as *Pseudomonas* and *Sphingomonas*, have been identified as key players in promoting plant growth and disease resistance (Pandey et al, 2023). The plant immune system actively shapes its microbiome by producing exudates that attract beneficial bacteria while repelling pathogens (Pascale et al, 2020). Furthermore, experimental evidence suggests that genetic variation can alter microbial community structure, thereby affecting plant fitness (Brachi et al, 2022). For instance, genetic loci involved in reactive oxygen species (ROS) production are known to regulate the leaf microbiome, highlighting the impact of host genetics on microbial communities (Pfeilmeier et al, 2021).

While the microbiome associated with roots and shoots are extensively studied, seed microbiome studies are not so advanced. Seeds, as primary agents of plant reproduction, encapsulate the evolutionary legacy of their parent plants and act as crucial interfaces between plant generations and their environment. The seed microbiome plays a key role in various aspects of plant physiology, including growth, phenology, and stress responses, which are vital for maintaining plant health and enhancing productivity (Berg and Raaijmakers, 2018; Hu et al, 2024; Wassermann et al, 2021). Therefore, studying the seed microbiome provides insights into plant-microbe co-existence and co-evolution and the potential for improving agricultural practices through microbiome management.

However, the mechanisms underlying seed microbiome assembly and function remain largely unexplored. Recent advances in seed microbiome research have demonstrated that both environmental conditions and host genotype shape the microbial communities within seeds. It is generally accepted that both maternal (the microbiome associated with the flower), paternal (pollen), and external factors (air/insect interactions during pollination) influence the composition of the seed microbiome

[1]Desert Research Initiative, Biological and Environmental Science and Engineering Division, 4700 King Abdullah University of Sciences and Technology, 23955-6900 Thuwal, Saudi Arabia. [2]Inst. of Ecological Microbiology, University Bayreuth, 95448 Bayreuth, Germany. [3]Department of Biological Sciences, College of Science, University of Jeddah, 80327 Jeddah, Saudi Arabia. [4]These authors contributed equally: Sabiha Parween, Naheed Tabassum. ✉E-mail: heribert.hirt@kaust.edu.sa

(War et al, 2023). For instance, previous studies on fonio millet revealed a diverse seed microbiome, with microbial composition correlating strongly with environmental variables and being associated with specific host genotypes (Tabassum et al, 2024). In this study, host genome SNPs were identified to be linked to genes involved in critical processes such as embryo/seed development and stress/defense responses, suggesting a genetic basis for seed microbiome composition and function. Therefore, the study of the seed microbiome and the genetic interactions with the host genome could have implications for plant breeding and sustainable agriculture practices. While crop plant research has advanced significantly, the model plant *Arabidopsis thaliana* offers crucial advantages for studying the seed microbiome due to its genetic diversity (represented by multiple ecotypes), which facilitates genome-wide association studies (GWAS). Furthermore, Arabidopsis offers genetic resources such as T-DNA lines for functional validation, along with a well-characterized genome, an extensive database, and a short life cycle, making it ideal for studying the relationship between the genome, the native microbiome, and the environment as well as exploring the mechanisms influencing seed-microbiome assembly.

In this study, we aimed to identify the key bacterial taxa in different Arabidopsis ecotypes and whether these taxa drive the assembly of the seed microbiome across distinct accessions. In addition, we investigated whether host genotype factors influence the microbiome diversity. Our findings indicate that the abundance of specific bacterial taxa is linked to the geographical and soil progeny of the ecotypes. Furthermore, we provide evidence that distinct bacterial taxa might drive the beta diversity and are associated with specific host genotype factors such as the RNA-binding protein RBP47B, which is involved in RNA condensate regulation, a post-translational process crucial for plant signaling and development (Xie et al, 2023). Overall, understanding how genotype factors regulate the seed microbiome diversity through RNA turnover may unveil novel insights how microbial communities are assembled in specific seed ecotpyes.

## Results

### Geographical location influences seed microbiome diversity of Arabidopsis

To study seed microbiome diversity in different ecotypes of *Arabidopsis thaliana*, we analyzed the seed microbiome of 227 accessions, each represented by four replicates. Accession records are reported at the country level, all analyses were performed using countries as the spatial unit, acknowledging that these represent coarse but standardized bins in biogeographic studies. Of these, 219 accessions originated from 24 countries distributed across Europe, North America, and Asia, reflecting diverse geographic origins (Fig. 1A). Sweden contributed the highest number of samples ($n = 47$), followed by France ($n = 44$), the United Kingdom ($n = 38$), and Germany ($n = 23$). Moderate contributions came from the Czech Republic ($n = 19$) and the United States ($n = 16$), while countries such as Italy, Russia, and others contributed between 1 and 4 samples each (Fig. 1B, Dataset EV1). The wide distribution of accessions ensures the robustness of our dataset, providing a strong foundation for global and cross-regional analyses.

We generated a total of 183,256,199 paired-end sequences targeting microbial 16S rRNA and after merging and quality filtering, 153,015,149 high-quality bacterial reads were retained, resulting in 30,053 non-singleton amplicon sequence variants (ASVs) with an average length of 377 bp (Dataset EV2A). To evaluate richness across accessions, rarefaction curves were generated using the Chao1 diversity index, showing adequate sequencing depth (Fig EV1, Dataset EV2B).

Alpha diversity, measured by the Shannon diversity index, varied significantly in accessions from different countries (Fig EV2). Geographic mapping revealed regional differences in richness and evenness of seed microbiota (Fig. 1C). Based on Shannon diversity, accessions were categorized into three groups: high ($\geq 5$), medium (3–4.99), and low (<3) diversity (Dataset EV1). Analysis of variance (ANOVA) indicate statistically significant differences among these categories. For example accessions from Sweden exhibited moderate diversity (range: 0.311–6.127, mean: 3.9) (Dataset EV1), whereas France, despite with fewer accessions, showed broader variation (range: 0.879–7.063, mean: 4.3). Moderate contributors such as the Czech Republic (range: 0.415–6.404, mean: 3.9) and the United Kingdom (range: 0.821–5.461, mean: 3.7) also displayed significant diversity. Interestingly, countries with minimal accession numbers, such as Pakistan and Finland, exhibited high Shannon diversity indices, suggesting that seed microbiome diversity in Arabidopsis is highly influenced by geographic and/or environmental factors. These findings highlight the diverse seed microbiome across Arabidopsis accessions, highlighting its potential role in local adaptation and ecological interactions.

### Specific bacterial taxa dominate the Arabidopsis seed microbiome but certain taxa are exclusively associated with particular accessions

Analysis of the relative abundance of bacterial communities in Arabidopsis identified the top ten genera as *Paenibacillus*, *Chryseobacterium Delftia*, *Brevibacillus*, *Cohnella*, *Stenotrophomonas*, *Pseudomonas*, *Burkholderia*, *Exiguobacterium* and one as unclassified genera (Figs. 2A and EV2A). However, their distribution varied across accessions coverage and total relativity (Fig. 2A,B). For instance, *Paenibacillus* was identified as the most dominant genus, with a relative abundance of 14.98, present in 74.42% of accessions across 19 countries. This was followed by *Chryseobacterium* (10.32) and *Delftia* (8.89), both consistently also found in 17 countries, reflecting their widespread distribution and ecological significance (Fig. 2A,B, Dataset EV1). Among the moderately abundant genera, *Cohnella* (2.52) and *Brevibacillus* (2.87) were detected in 12 countries, indicating a more region-specific distribution. Similarly, *Stenotrophomonas* (2.46) was broadly distributed across 18 countries and detected in 53.42% of accessions, suggesting its stability despite its lower abundance. *Pseudomonas* (1.56) was present in 15 countries, reflecting a more localized but ecologically relevant presence, whereas *Burkholderia* and *Exiguobacterium* (both with a relative abundance of 0.37), were each detected in only 7 countries. Interestingly, we did not find a common genus in all countries, indicating the lack of a core seed microbiome across all geographical locations (Fig. EV2B). Instead, we observed a number of unique genera in a significant number of countries, suggesting niche-specific ecological roles or adaptations of specific genera to specialized environments. Our results

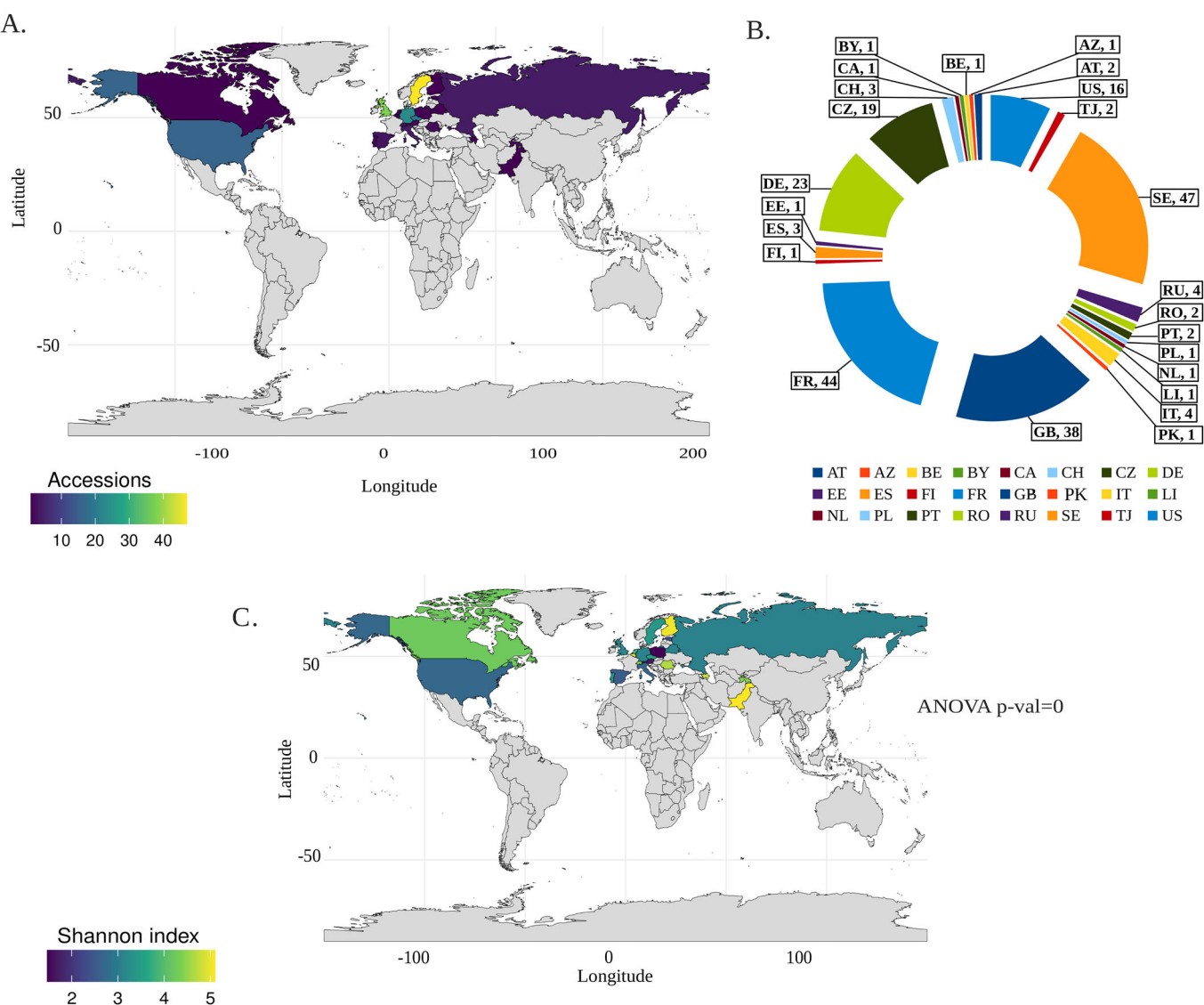

**Figure 1. Geographic distribution of seed microbiome alpha diversity of Arabidopsis accessions.**

(A) Distribution across 24 countries on a world map, with the frequency of 219 accessions highlighted using a heatmap gradient where lighter shades indicate higher accession counts. Gray regions represent countries with no recorded accessions. Major latitude and longitude grid lines are overlaid to provide spatial context. (B) Pie chart showing the proportional distribution of accessions by country using ISO 3166-1 alpha-2 country codes. AT: Austria, AZ: Azerbaijan, BE: Belgium, BY: Belarus, CA: Canada, CH: Switzerland, CZ: Czech Republic, DE: Germany, EE: Estonia, ES: Spain, FI: Finland, FR: France, GB: United Kingdom (Great Britain), IND: India, IT: Italy, LI: Liechtenstein, NL: Netherlands, PL: Poland, PT: Portugal, RO: Romania, RU: Russia, SE: Sweden, TJ: Tajikistan, US: United States. (C) Map of average Shannon diversity index values by country, shown as a heatmap gradient (light = high, dark = low). Diversity was categorized as High (≥5), Medium (3–4.99), or Low (<3), with countries lacking data shown in gray. Source data are available online for this figure.

demonstrate the dominance of several genera, the variability in distribution of moderately abundant taxa, and the rarity of certain groups. This result provides a basis for understanding microbial contributions to host plant adaptation and resilience in different environments.

## Key bacterial taxa regulate Arabidopsis seed microbiome diversity

To identify the key microbiome members that shape overall seed microbiome diversity, we determined β-diversity of the seed microbiome using both Weighted UniFrac and Jaccard metrics. Principal coordinate analysis (PCoA) revealed that PCoA1 plus PCoA2 explained 10.17% of the variance in Jaccard β-diversity and 50.25% in Weighted UniFrac β-diversity (Fig. EV3). To explore the relationship between taxon abundance and β-diversity, we included all genera with a relative abundance of ≥0.1% and evaluated both correlation strength and abundance distributions (Fig. 3). *Delftia, Stenotrophomonas, Chryseobacterium, Paenibacillus* and *Cohnella* showed significant correlations with both β-diversity metrics, though with contrasting directions (Fig. 3A,B). Notably, in a large number of accessions, *Delftia, Stenotrophomonas* and

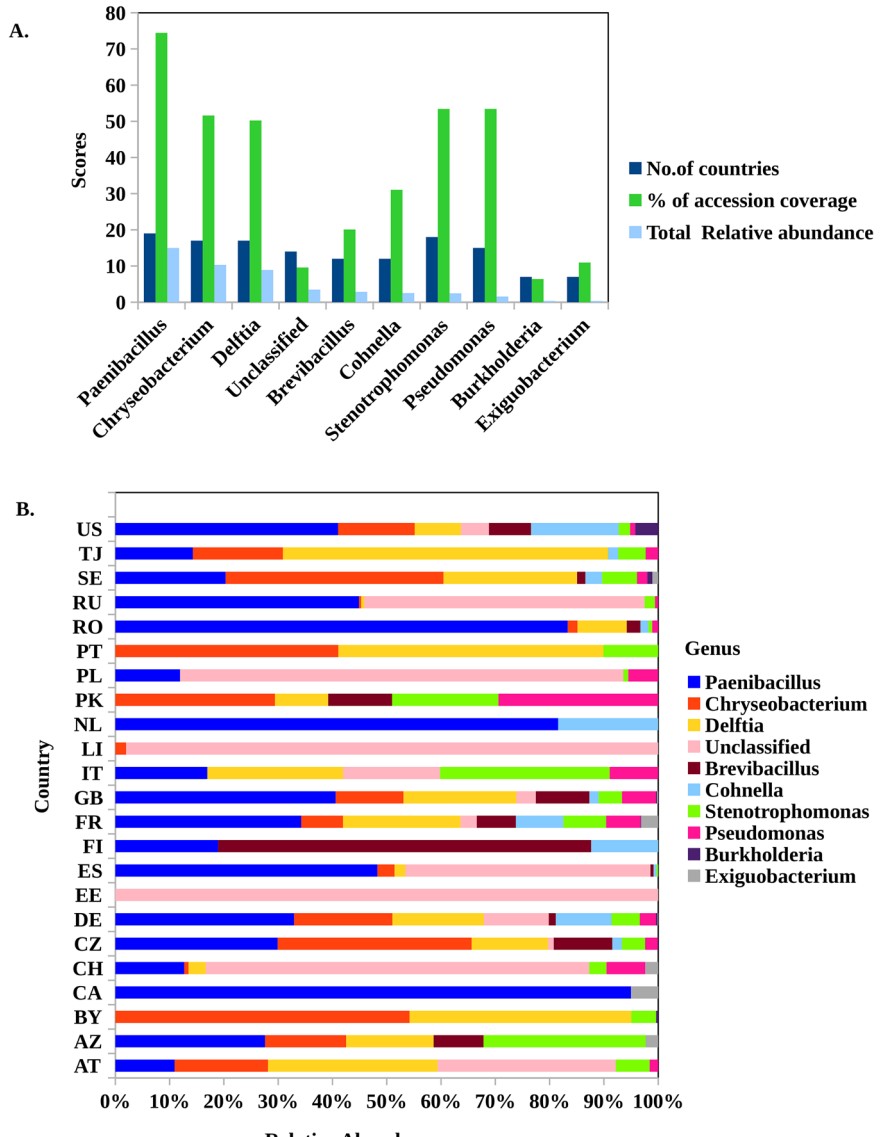

**Figure 2. The relative abundance of top 10 genera.**

(A) Relative abundance, percentage of accessions covered, and the number of countries for the top 10 bacterial genera. (B) The distribution of each genus across countries. Each genus is highlighted by a different color. The country corresponds to AT: Austria, AZ: Azerbaijan, BE: Belgium, BY: Belarus, CA: Canada, CH: Switzerland, CZ: Czech Republic, DE: Germany, EE: E.stonia, ES: Spain, FI: Finland, FR: France, GB: Great Britain, PK: Pakistan, IT: Italy, LI: Liechtenstein, NL: Netherlands, PL: Poland, PT: Portugal, RO: Romania, RU: Russia, SE: Sweden, TJ: Tajikistan, US: United States. Source data are available online for this figure.

*Chryseobacterium* exhibited strong positive correlations with Weighted UniFrac (Fig. 3B). In contrast, accessions lacking these genera were dominated by *Paenibacillus*, and to a lesser *extent Cohnella, Brevibacillus, Salinisphaera* and *Mycoplasma* (Fig. 3C–F). Co-occurrence analysis further revealed strong positive associations among *Delftia, Stenotrophomonas* and *Chryseobacterium*, coupled with a negative correlation with *Paenibacillus* (Fig. EV4, Dataset EV3).

To validate and quantify these associations, we applied distance-based statistical tests. In envfit, *Delftia* ($R^2 = 0.916$, $p = 0.001$), *Chryseobacterium* ($R^2 = 0.383$, $p = 0.001$), and *Stenotrophomonas* ($R^2 = 0.394$, $p = 0.001$) were strongly correlated with ordination space, whereas *Paenibacillus* and *Cohnella* showed only weak

effects. PERMANOVA/adonis2 identified *Delftia* as the dominant driver of β-diversity ($R^2 = 0.39$, $p = 0.001$), with *Stenotrophomonas* contributing significantly ($R^2 = 0.0099$, $p = 0.002$), while *Chryseobacterium* showed no independent contribution after accounting for collinearity. dbRDA constrained ordinations confirmed *Delftia* ($F = 31.3$, $p < 0.001$), *Chryseobacterium* ($F = 12.4$, $p < 0.001$), and *Stenotrophomonas* ($F = 12.7$, $p < 0.001$) as the strongest explanatory taxa, with *Paenibacillus* and *Cohnella* only marginal (Dataset EV7).

Together, these results demonstrate that *Delftia, Stenotrophomonas* and *Chryseobacterium* consistently emerge as a "triad" of key taxa shaping seed microbiome structure. Their abundance correlates strongly with phylogenetic dissimilarities captured by Weighted UniFrac, while their absence is linked to communities

**A.** **Distance:Jaccard**

| Genus | Corr,r | P value |
|---|---|---|
| Delftia | -0.55 | 1.2E-17 |
| Chryseobacterium | -0.51 | 1.1E-14 |
| Stenotrophomonas | -0.52 | 2.6E-15 |
| Escherichia | -0.2 | 0.01 |
| Staphylococcus | -0.21 | 0.02 |
| Ralstonia | -0.19 | 0.04 |
| Unclassified_200 | -0.19 | 0.04 |
| Paenibacillus | 0.41 | 3.8E-09 |
| Cohnella | 0.32 | 2.8E-05 |
| Brevibacillus | 0.26 | 0.001 |

**B.** **Distance: Weighted Unifrac**

| Genus | Corr,r | P value |
|---|---|---|
| Delftia | 0.69 | 1.2E-31 |
| Chryseobacterium | 0.67 | 6.4E-30 |
| Stenotrophomonas | 0.72 | 1.6E-35 |
| Paenibacillus | -0.32 | 1.4E-05 |
| Salinisphaera | -0.30 | 0.0001 |
| Mycoplasma | -0.30 | 0.0001 |
| Unclassified_201 | -0.29 | 0.0001 |

**C.** 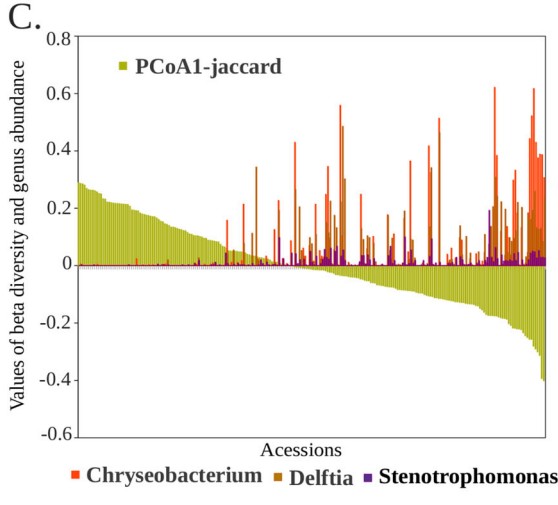

**D.** 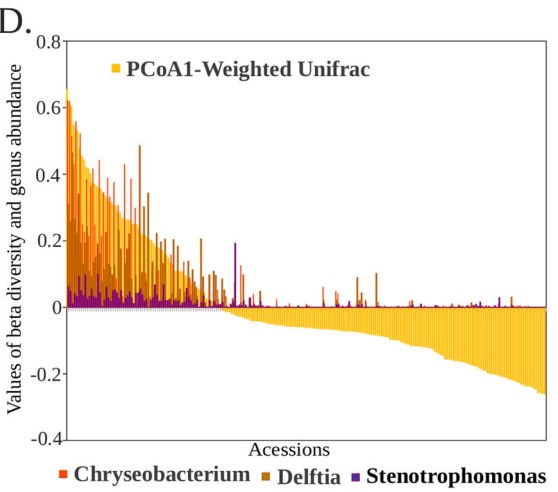

**E.** 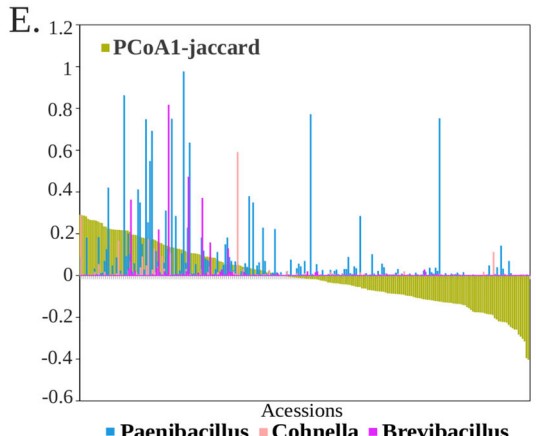

**F.** 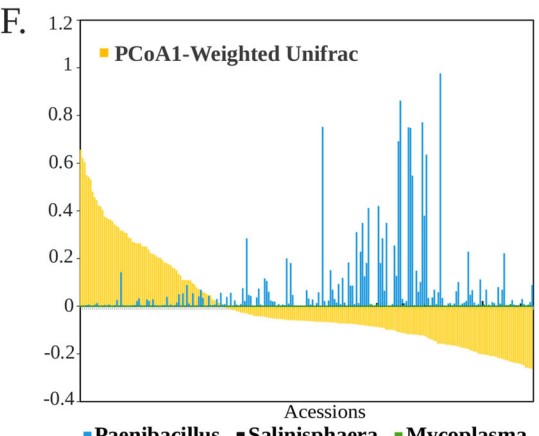

Figure 3.   Correlation and abundance patterns of key genera in the seed microbiome.

(A, B) Tables displaying significant correlation between genera and the two diversity measures, Jaccard and Weighted UniFrac, respectively. (C, D) Graphs showing the relative abundance trends of the key genera *Delftia, Stenotrophomonas* and *Chryseobacterium* across all Arabidopsis accessions in relation to the PCoA1 values derived from Jaccard and Weighted UniFrac analyses. Genera and diversity measures are represented using distinct colors for clarity. Weighted UniFrac incorporates both phylogenetic relationships and taxon abundance, whereas Jaccard beta diversity considers only the presence or absence of taxa, disregarding their abundance. (E) Relative abundance trend of *Paenibacillus, Cohnella* and *Brevibacillus* with PCoA1 of Jaccard and (F) of *Paenibacillus, Salinisphaera* and *Mycoplasma* with PCoA1 of Weighted Unifrac distance measure, respectively. Source data are available online for this figure.

dominated by *Paenibacillus* and related taxa. The convergence of correlation-based and distance-based methods underscores their unique role as central drivers of microbial turnover across seed accessions.

## Certain taxa share common functions related to iron metabolism or hemicellulose degradation

Given the significant correlations observed among *Delftia, Chryseobacterium* and *Stenotrophomonas* (Triad 1, Fig. EV4) and *Paenibacillus, Cohnella* and *Brevibacillus* (Triad 2, Fig. EV4), we investigated whether these triads share common functional attributes. Using the PICRUSt software, we first checked the sample-level weighted NSTI (Nearest Sequenced Taxon Index) which is average NSTI, weighted by their relative abundance which falls into a good prediction reliability of ~0.01–0.15 (Appendix Fig. S2, Dataset EV6). We then specifically focused on ASVs belonging to the Triad 1 and Triad 2 groups, previously identified as having the strongest influence on β diversity, thereby reducing noise from unrelated taxa and improving interpretability. We ranked and selected the top 20 KEGG Orthologs by predicted copy number, which reflects the dominant and most reliable functional potential. Importantly, functional inference at the copy number level avoids the pitfall of relying solely on taxon abundance and this approach is widely accepted in predictive metagenomics. From the selected KEGG Orthologous (KOs), we plotted Venn diagrams (Fig. 4A,B and Dataset EV5). For Triad 1, we found three KEGG annotations that were common in *Delftia-Chryseobacterium-Stenotrophomonas* (Fig. 4A), encoding for TC,FEV,OM, an iron complex outer membrane receptor protein, which is integral to bacterial iron acquisition and transport mechanisms that is critical for bacterial survival. Moreover, we identified an RNA polymerase sigma-70 factor (ECF subfamily), which regulates bacterial responses to environmental stress and an ABC-2 type transport system permease.

Although *Paenibacillus* was less strongly correlated with the taxa of *Cohnella* and *Brevibacillus* in a large number of accessions (Figs. EV4 and 3A,C), we also analyzed the KEGG IDs encoded by the ASVs that overlapped for this triad (Fig. 4B). Interestingly, a overlap was obtained for functions in hemicellulose degradation and sugar transport of these taxa. Hemicelluloses are essential compounds of plant cell walls, accounting for about a quarter of all plant biomass worldwide. However, in seed cotyledons and the endosperm of species from several plant families, hemicelluloses can also be used as mobile carbon reserves (Hoch, 2007).

Overall, these data suggest that specific accessions are associated with particular groups of bacterial genera, and that the distinct functional profiles of these groups may contribute to or reflect adaptation to their respective environments.

## GWAS identifies host genetic loci associated with the common functions of *Delftia, Chryseobacterium* and *Stenotrophomonas*

To test whether any of the KO values of the two triads were linked to the host genotype, we performed genome-wide association studies (GWAS) to these common KO abundance values. For Triad 1, we obtained a number of hits for both K02014 and K03088 (Fig. 5A, Dataset EV4). For the gene locus AT3G19130, encoding the RNA-binding protein 47B (RBP47B), 2 SNPs were found for K02014 and 1 SNP for K03088, all of which were in the promoter or 5' UTR region of the RBP47 gene (Fig. 5B). The consistent occurrence of RBP47B across the common KO categories prompted us to further investigate its potential association with the key triad. To explore this relationship, the relative abundance values of the key genera of the triad were used as phenotype in GWAS. This analysis revealed a notable association between *Chryseobacterium* and *RBP47* as a set of single-nucleotide polymorphisms (SNPs), located at the AT3G19130 locus (Fig. 5A). All the SNPs at the AT3G19130 locus were identified to have a modifier effect on the associated KO categories, suggesting that the RNA-binding protein 47B might play an important role related to the Arabidopsis seed microbiome assembly.

### *RBP47B* affects the relative abundance of taxa in Col-0

Since *RBP47B* was identified as a common target across the key genera and KO categories, we next investigated whether *RBP47B* might also affect the Arabidopsis seed microbiome composition. *RBP47B* belongs to a larger gene family and quadruple *rbp47abcc'* mutants in the Col-0 accession have been instructive as to the function of *RBP47* in Arabidopsis. For this reason, we compared the microbiomes of Arabidopsis Col-0 WT with the Col-0 *rbp47abcc'* mutant. To ensure comparability, WT and *rbp47abcc'* mutant were propagated on the same soil as described previously (Tabassum et al, 2024) for all accessions. To avoid contamination from epiphytic soil microbes, seeds were surface sterilized and extensively washed before germination in sterile conditions for 2 days. After DNA extraction and microbiome analysis of four replicates, the relative abundance of WT and *rbp47abcc'* was analyzed. The stack plot highlights significant shifts in the family proportions upon mutation of *rbp47abcc'* (Fig. 6A), with *rbp47abcc'* mutants showing pronounced changes in the abundance of *Rhizobiaceae* and Burkholderiaceae. These results suggest that mutation of *rbp47abcc'* alters the microbial niche to favor certain bacterial families over others. Moreover, the seed microbiome of *rbp47abcc'* mutant contained no taxa of *Delftia* and *Chryseobacterium* and strongly reduced *Stenotrophomonas*, indicating that *rbp47abcc'* mutant shapes the seed microbiome of Arabidopsis.

A.

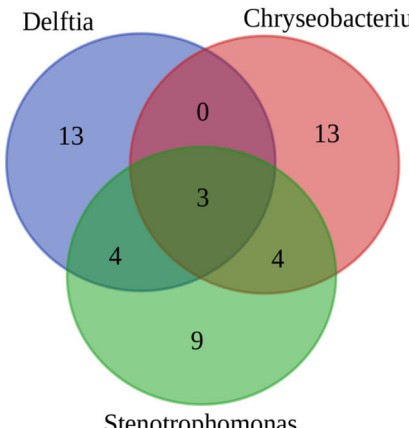

| Triad | KO ID | Description |
|---|---|---|
| Chryseobacterium, Delftia, Stenotrophomonas | **K01992** | ABC-2.P; ABC-2 type transport system permease protein |
| | **K02014** | TC.FEV.OM; iron complex outermembrane recepter protein |
| | **K03088** | rpoE; RNA polymerase sigma-70 factor, ECF subfamily |

B.

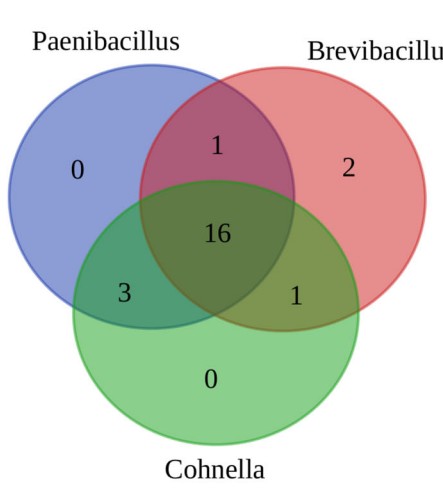

| Triad | KO ID | Description |
|---|---|---|
| Paenibacillus ,Brevibacillus, Cohnella | K17318 | K17318, lplA; putative aldouronate transport system substrate-binding protein |
| | K02026 | ABC.MS.P1; multiple sugar transport system permease protein |
| | K06147 | ABCB-BAC; ATP-binding cassette, subfamily B, bacterial |
| | K01992 | ABC-2.P; ABC-2 type transport system permease protein |
| | K01990 | ABC-2.A; ABC-2 type transport system ATP-binding protein |
| | K02529 | lacI, galR; LacI family transcriptional regulator |
| | K02025 | ABC.MS.P; multiple sugar transport system permease protein |
| | K17319 | lplB; putative aldouronate transport system permease protein |
| | K17320 | lplC; putative aldouronate transport system permease protein |
| | K03406 | mcp; methyl-accepting chemotaxis protein |
| | K02027 | ABC.MS.S; multiple sugar transport system substrate-binding protein |
| | K00059 | fabG; 3-oxoacyl-[acyl-carrier protein] reductase [EC:1.1.1.100] |
| | K03088 | rpoE; RNA polymerase sigma-70 factor, ECF subfamily |
| | K02015 | ABC.FEV.P; iron complex transport system permease protein |
| | K07718 | yesM; two-component system, sensor histidine kinase YesM [EC:2.7.13.3] |
| | K07720 | yesN; two-component system, response regulator YesN |

**Figure 4. Functional analysis of triads.**

Venn diagram depicting the top 20 KEGG Orthologs (KOs) associated with (**A**) Triad 1 (*Delftia-Chryseobacterium- Stenotrophomonas*) and (**B**) Triad 2 (*Paenibacillus-Cohnella-Brevibacillus*), identified using PICRUSt-based functional predictions with the description of overlapping KEGG Orthologs shared between the two triads. Source data are available online for this figure.

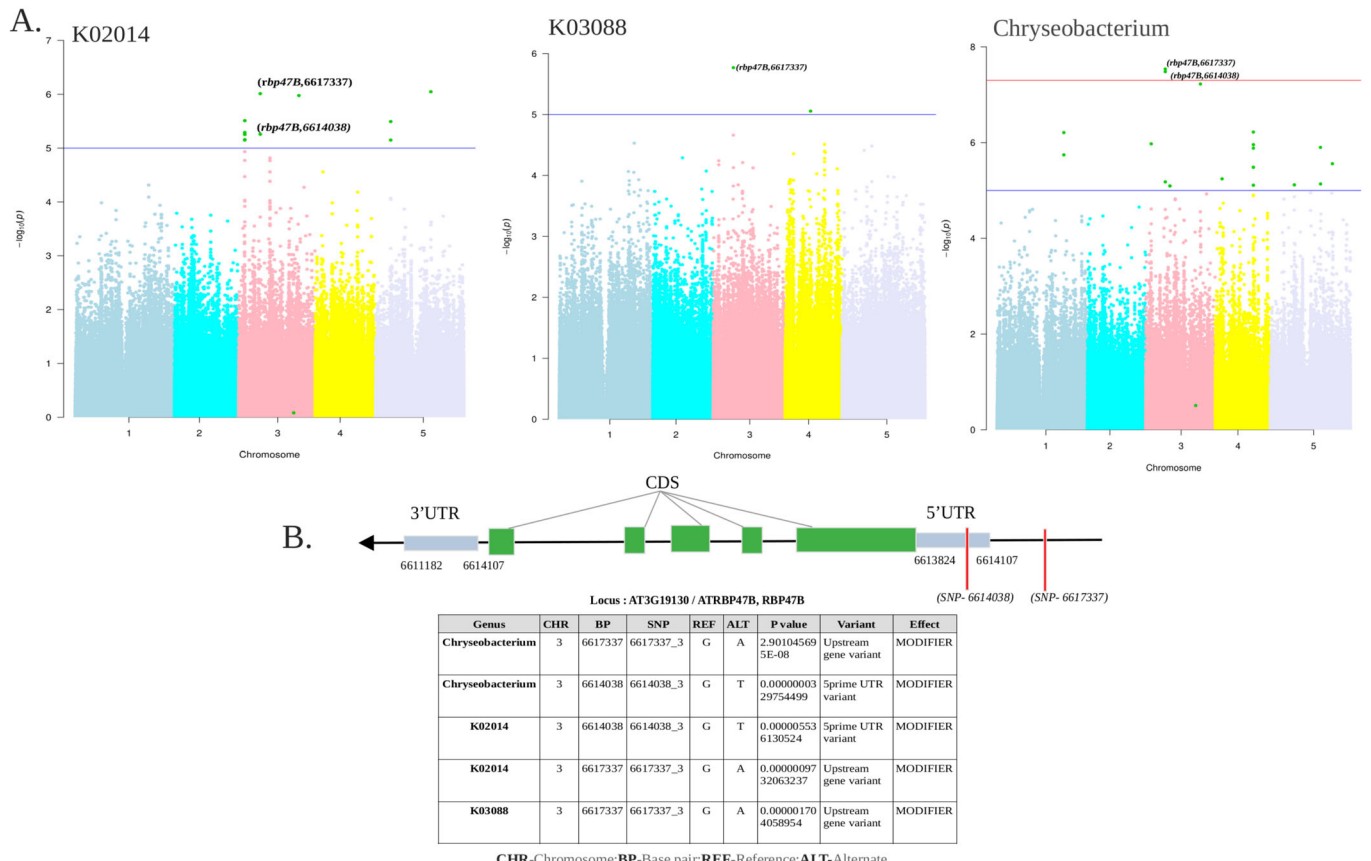

**Figure 5. Genome-wide associations of Arabidopsis loci with triad functional traits.**

(A) Manhattan plot highlighting the SNPs associated with the locus encoding the *RBP47B* gene, which is shared by the common KOs of Triad 1 (*Delftia-Chrysobacterium-Stenotrophomonas*) and the relative abundance of the genus *Chryseobacterium*. The plot includes two horizontal lines indicating *p*-value thresholds, with SNPs passing the suggestive threshold (*p*-value < 0.05) marked in green. (B) Shows the SNP positions and their effects on the genes.

### RBP47B modulates plant tolerance to low pH and Iron toxicity in association with the bacterial triad

Previous studies demonstrated that beneficial plant-associated microbes contribute to host stress adaptation by modulating nutrient uptake, pH homeostasis, and metal detoxification pathways (Bakker et al, 2018; Harbort et al, 2020; Msimbira and Smith, 2020). The enhanced abundance of *Delftia, Stenotrophomonas* and *Chryseobacterium* in specific countries (Fig. EV5) motivated us to search for environmental, geophysical and/or meteorological factors that might be involved in shaping the association between the three bacterial taxa with these Arabidopsis accessions. Two factors emerging from this search was an overlap of the location of these accessions to mining sites with high iron contents and at the same time soils of low pH (Appendix Fig. S1). Low pH in the presence of iron can lead to iron toxicity posing a major challenge to plant growth due to increased $Fe^{3+}$ solubility, resulting in oxidative stress and nutrient imbalance (Connolly and Guerinot, 2002; Vélez-Bermúdez and Schmidt, 2023). To test whether *RBP47* might be involved in the adaptation of the respective Arabidopsis accessions to iron toxicity, we investigated whether *rbp47abcc'* mutant plants exhibited altered sensitivity to iron toxicity at low pH. We therefore conducted phenotypic experiments under three pH conditions (pH 5.8, pH 4.0, and pH 3.5) and an additional iron-supplemented condition (pH 4.0 + 28 mg/L Fe). In wild-type plants, growth was reduced by 59% at pH 4.0 and 85% at pH 3.5 compared to pH 5.8 (Fig. 6A,B). However, these reductions were more pronounced in *rbp47abcc'* mutants, which compared to WT, showing an additional reduction of fresh weight by 15.79% at pH 4.0, 45.16% at pH 4.0 + Fe, and 34.75% at pH 3.5 (Fig. 6B,C). These findings suggest that *RBP47* is essential in helping Arabidopsis to adapt to the soil conditions of high iron and low pH (Appendix Fig. S1). At the same time, *RBP47* is also involved in shaping the microbiome of these Arabidopsis accessions (Fig. 7).

## Discussion

Plants constantly balance their growth and stress adaptations via multiple interconnected signaling modules. Recent studies have demonstrated that the plant microbiome plays a crucial role in regulating the host plant growth and overall health, together being denoted as holobiont (Vandenkoornhuyse et al, 2015). A fundamental part of the plant microbiome is the seed microbiome which is known to be affected by host genotypes and geographical

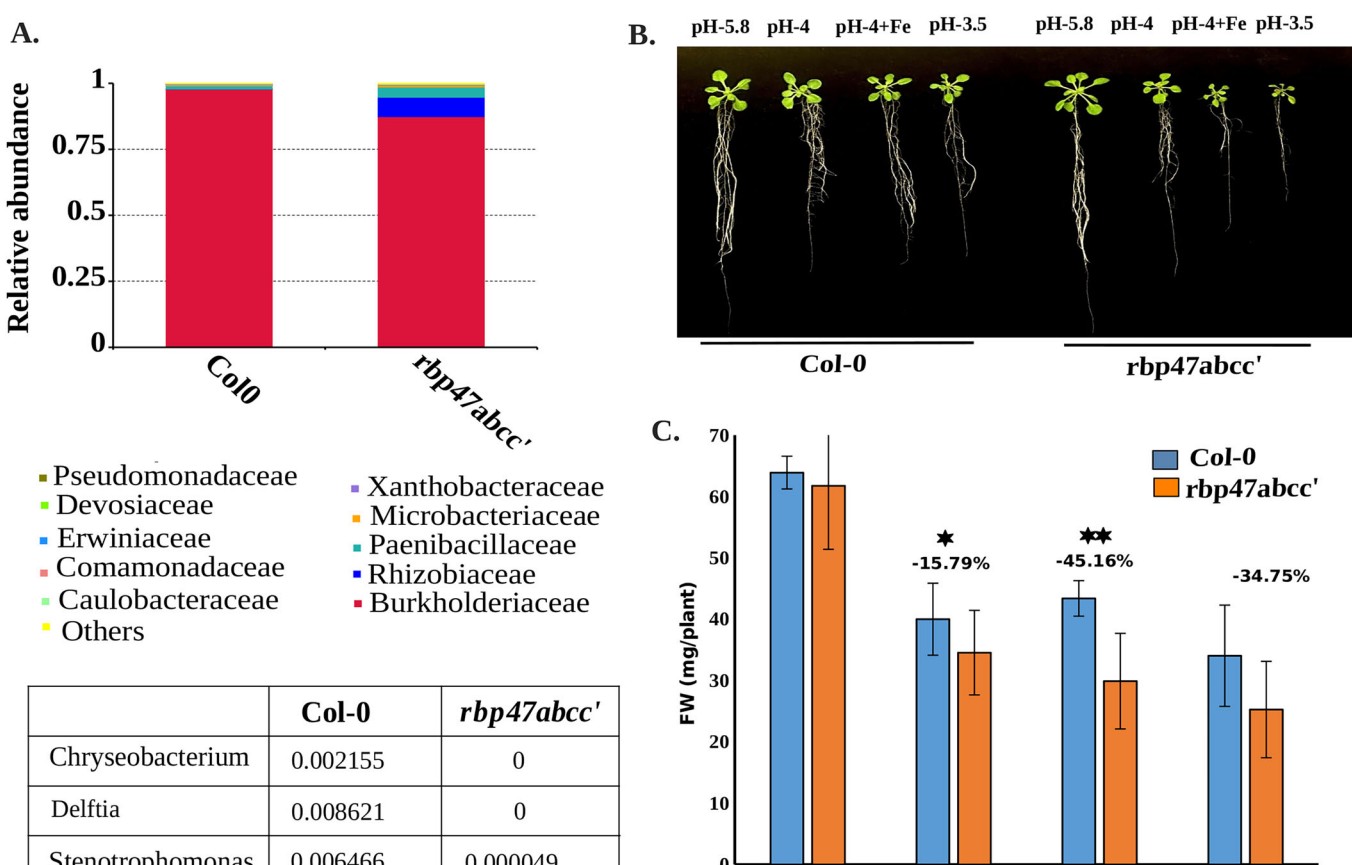

**Figure 6. rbp47abcc′ affects microbial abundance and iron toxicity.**

(A) Relative abundance of microbial taxa at the family level in WT and *rbp47abcc′* mutant. Each stack represents the proportional contribution of different microbial families within the total microbial community for each sample type. Differences in relative abundance between WT and the *rbp47abcc′* mutant are visualized as variations in the percentage area occupied by each family in the stacks. (B) Representative images of WT and *rbp47abcc′* mutant plants grown under different pH conditions (5.8, 4.0, and 3.5) and iron toxicity stress (pH 4.0 + 28 mg/L Fe) for 15 days. (C) Quantification of fresh weight showing relative growth reduction in WT and *rbp47abcc′* mutants under each condition. Error bars represent standard deviation (SD) calculated from three biological replicates, comprising a total of 25 plants per treatment ($n = 25$). Statistical significance was determined using Student t-test with P-value 0.019 (*) and P-value 0.0031 (**). Source data are available online for this figure.

locations. To understand whether microbial taxa govern the overall seed microbiome, we studied the Arabidopsis seed microbiome and identified genera such as *Paenibacillus*, *Delftia*, *Stenotrophomonas*, and *Chryseobacterium* that were widespread across geographic locations. Interestingly, a meta-analysis of 50 plant species (including Arabidopsis) from diverse regions worldwide revealed that seed microbiota are highly diverse and include both a variable and a stable common core fraction (Simonin et al, 2022). The common bacterial taxa identified across samples include *Pseudomonas*, *Pantoea*, *Rhizobium*, *Sphingomonas*, *Methylobacterium*, and *Paenibacillus*. While *Paenibacillus* was among the most abundant taxa in a range of Arabidopsis accessions, many accessions were completely devoid of this taxon. In contrast, *Pseudomonas* was present in almost all accessions but only at relatively low abundance levels. We cannot exclude, however, that certain unclassified taxa may correspond to the missing *Pseudomonas* taxon in these accessions. Interestingly, we noted that several accessions showed a clear pattern of co-occurrence of several bacterial taxa, notably identifying *Delftia*, *Stenotrophomonas* and *Chryseobacterium* as a triad. We also found that accessions, which

were dominated by *Paenibacillus*, were associated with *Cohnella* and *Brevibacillus* taxa. Interestingly, the large majority of these accessions showed no presence of *Delftia*, *Stenotrophomonas* and *Chryseobacterium*, suggesting that distinct accessions possess very different functional networks. These findings support the concept that direct bacteria–bacteria interactions can be responsible for population shifts in microbiomes (Schäfer et al, 2022). In the framework of the concept of "guild by association," microbial communities are often organized into functional or ecological groups in the same environment (Simberloff and Dayan, 1991). This concept emphasizes that different bacterial species may share molecular mechanisms that enable them to collaborate to fulfill specific functional roles (Wu et al, 2021). Applying this concept to the co-occurrence of the *Delftia-Stenotrophomonas-Chryseobacterium* triad, we identified two functional KO terms which are related to the iron metabolism of the bacteria. The *Paenibacillus-Cohnella-Brevibacillus* triad shared KO terms of hemicellulose degradation and sugar transport. Hemicelluloses make up a large part of wood but are also storage molecules of seeds that are released during germination. Bacteria that can metabolize these components will

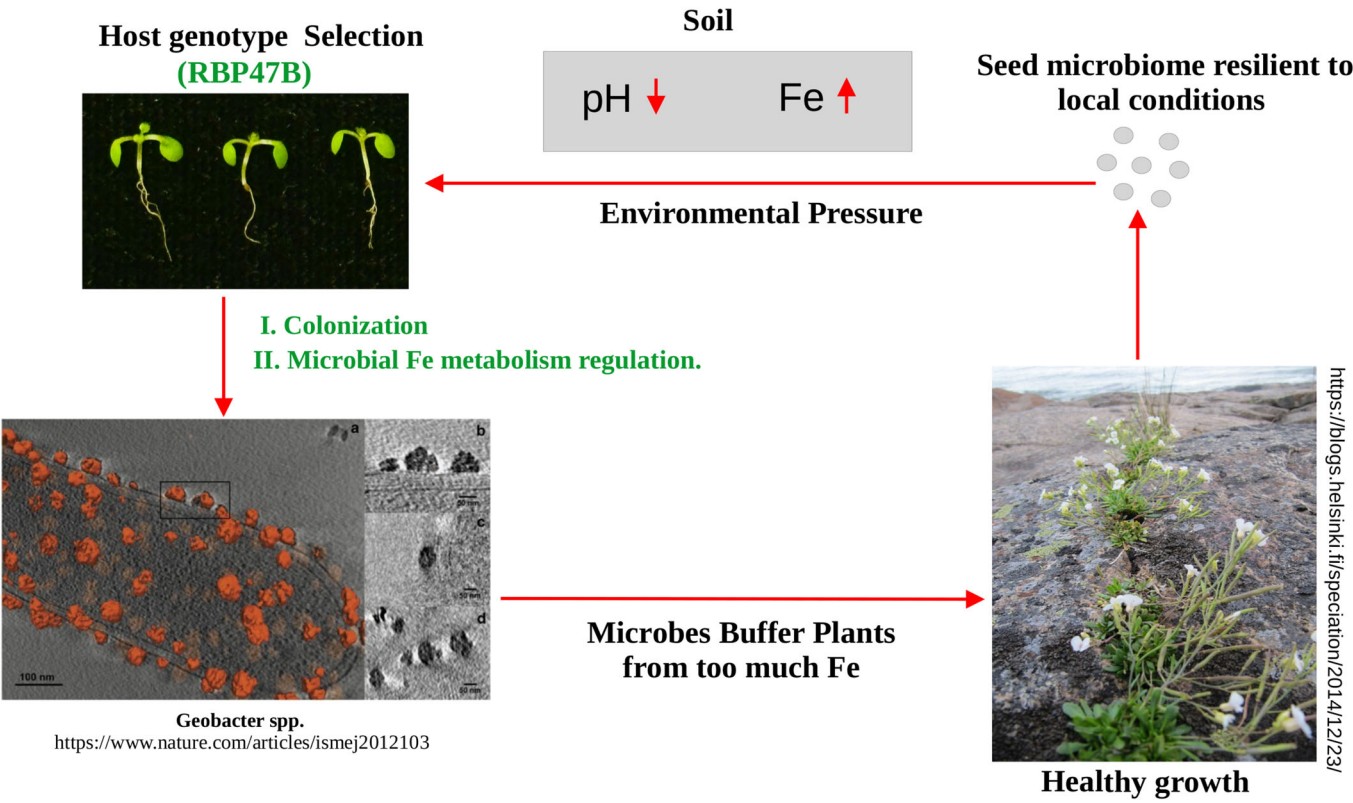

**Figure 7. Proposed mechanism of Arabidopsis seed microbiome assembly.**

The host genotype is influenced by geographical location, with stress-associated proteins like RBP47 potentially altering RNA turnover in response to internal or external signals. The host-specific *RBP47* genotype may shape the seed microbiome and thereby also help adaptation to the soil and geo-climatic conditions.

have an advantage in colonization of the host plant (Yang et al, 2021). It is plausible to assume that *Paenibacillus, Cohnella* and *Brevibacillus* specialized in this function for ensuring their dominance in these accessions but further investigations are necessary to clarify this hypothesis.

The significance of the exclusive co-occurrence of the bacterial triad *Delftia-Stenotrophomonas-Chryseobacterium* was further supported by mapping the geographical distribution of the respective accessions, showing a strong association with certain countries such as Sweden, Germany and France. However, it should be noted that in these countries, we also found a number of accessions which did not show the high abundance of this bacterial triad. These results suggest that neither latitude nor climatic conditions could be responsible for the particular microbial distribution of taxa in these accessions. This shifted our focus to the possibility that soil-specific features might be drivers for the occurrence of the bacterial triad. In fact, we found that the soil at the sites of the accessions showed a low pH and often correlated with mining sites for iron, conditions which are known to be unfavorable for plant growth due to enhanced iron toxicity.

GWAS is a powerful tool for identifying genetic associations between the host genome and its associated microbiome (Deng et al, 2021). Our GWAS study using either of the triad bacterial taxa or the two KO terms indeed identified a number of genes but pointed to *RBP47B* as a common candidate. Our phenotypic assays showed that *rbp47abcc'* mutants exhibit significantly greater

sensitivity to acidic stress and excess iron compared to the wild type, suggesting that the *RBP47* gene family works as a key factor in plant stress resilience. Microbiome analysis of *rbp47abcc'* mutants further revealed the absence of the bacterial triad in these plants. These results suggest that *RBP47* family members may facilitate the recruitment or maintenance of a beneficial microbial community under acidic and iron-rich conditions. Iron toxicity at low pH poses a major challenge to plant growth due to increased $Fe^{3+}$ solubility, leading to oxidative stress and nutrient imbalances (Fang et al, 2024; Vélez-Bermúdez and Schmidt, 2023). The significantly higher sensitivity of *rbp47abcc'* mutants under iron-supplemented conditions (pH 4.0 + Fe) reveals that *RBP47* family members may play a role in iron homeostasis, potentially by mediating interactions with iron-mobilizing microbes or by regulating host-driven iron detoxification mechanisms. Interestingly, a recent study also reveals that a beneficial microbe specifically triggers the expression of translation-related genes (Yang et al, 2025). Moreover, a variety of ribosomal proteins and translation regulators are essential for the growth promotion mediated by beneficial microbes. This supports the robustness of our GWAS findings. In this context, it would be interesting to set up a SynCom (synthetic community) experiment to determine whether *RBP47B* (or the microbiome regulated by RBP proteins) has a broader role in enhancing root tolerance to stress. Additionally, it remains to be explored whether this tolerance extends to other soil metals. RBP47B was identified as an RNA-binding protein of stress granules (SG) in human and

Arabidopsis (Weber et al, 2008). SGs are dynamic, membrane-less organelles that form in response to environmental and internal stimuli and regulate post-transcriptional RNA metabolism (Alberti and Carra, 2018). Recently, the function of the Arabidopsis *RBP47* gene family has also been shown to mediate polyphenolic acid (PA)-induced translation (Xie et al, 2023). PAs are found in root exudates and are involved in chemical communication between plants and bacteria (Mandal et al, 2010). RBP47B was shown to directly bind to PAs and regulate PA-induced stress granules. Our functional Arabidopsis *rbp47abcc'* knockouts also provided direct evidence for a role of *RBP47* genes in shaping the microbiome composition and in plant resilience to acidic stress and iron toxicity. Future studies should clarify whether PAs are produced by Arabidopsis roots in response to these stress conditions and whether they influence the plant-microbiome interaction in a specific manner. Overall, this work opens promising avenues for engineering seed microbiomes to improve crop performance and ecological sustainability in challenging soil conditions.

# Methods

### Reagents and tools table

| Chemicals/reagents/resource | Reference/source | Catalogue number |
|---|---|---|
| Arabidopsis thaliana 227 ecotype seeds | Wolfgang Busch, Salk Institute, La Jolla, USA | |
| Arabidopsis thaliana Col-0 and rbp47abcc' seeds | Monika Chodasiewicz, KAUST, Thuwal, KSA | |
| Murashige and Skoog medium | Duchefa-biochemie | M0221 |
| 2-(N-morpholino) ethanesulfonic acid (MES) | Bioland Scientific LLC | CM01-500mg |
| Agar | Sigma-Aldrich | A1296 |
| Tris base | Millipore | 648310-M |
| Ethanol | Sigma-Aldrich | E7023 |
| Sodium Dodecyl Sulfate (SDS) | Sigma-Aldrich | L3771 |
| Sodium Chloride | SAFC | 1.064 |
| Ethylenediaminetetraacetic acid (EDTA) | Sigma-Aldrich | E4884 |
| EDTA-FeNa | Sigma-Aldrich | 03650-250G |
| Potassium hydroxide | Sigma-Aldrich | 221473 |
| **Others** | | |
| Seed Storage Chamber | LABEC Seed Storage Chambers | |
| Homogenizer | TissueLyser | |
| Growth chamber | Phytochamber/conviron | |
| **Software** | | |
| QIIME 2 [QIIME2 2022.2] | https://qiime2.org/ | RRID:SCR_021258 |
| Gephi [0.10.1] | https://gephi.org/ | RRID:SCR_004293 |
| PICRUSt2 [2.4.1] | https://github.com/picrust/picrust2 | RRID:SCR_022647 |

| Chemicals/reagents/resource | Reference/source | Catalogue number |
|---|---|---|
| PLINK [2.0] | https://www.cog-genomics.org/plink/ | RRID:SCR_001757 |
| EMMAX [2012021] | https://genome.sph.umich.edu/wiki/EMMAX | RRID:SCR_024012 |
| R [4.5.1] | https://www.r-project.org/ | RRID:SCR_001905 |
| **R packages** | | |
| Rnaturalearth [1.1.0] | https://CRAN.R-project.org/package=rnaturalearth | |
| Rnaturalearthdata [1.0.0] | https://CRAN.R-project.org/package=rnaturalearthdata | |
| Vegan [2.7.1] | https://CRAN.R-project.org/package=vegan | RRID:SCR_011950 |
| Phyloseq [1.52.0] | https://bioconductor.org/packages/phyloseq | RRID:SCR_013080 |
| vegan::envfit [2.7.1] | https://CRAN.R-project.org/package=vegan | RRID:SCR_011950 |
| vegan::adonis2 (PERMANOVA) [2.7.1] | https://CRAN.R-project.org/package=vegan | RRID:SCR_011950 |
| vegan::capscale (dbRDA) [2.7.1] | https://CRAN.R-project.org/package=vegan | RRID:SCR_011950 |

## Plant material and growth conditions

A seed collection from 227 Arabidopsis ecotypes from different geographical locations, including Europe, Asia and North America, was kindly provided by Wolfgang Busch, Salk Institute, La Jolla, USA. These accessions were part of the RegMap panel (Horton et al, 2012) and seeds were obtained by growing the parental generation of all lines used side by side in the same growth chambers under the same conditions (Meijón et al, 2014). To standardize the input of microbes from the soil, seeds were propagated for three generations on the same commercial soil (Sun Gro, Agraham, USA). The collection was propagated and maintained at the KAUST seed storage facility (Thuwal, KSA). For extraction of seed-associated microbial DNA, we followed the protocol of Tabassum et al (2024). Briefly, seeds were surface sterilized by shaking seeds with 70% ethanol + 0.05% sodium dodecyl sulfate for 10 min, followed by washing two times in 96% ethanol and air drying. Seeds were then germinated in sterile conditions on 1/2 MS (Murashige & Skoog) media for 2-3 days before freezing and pulverization in liquid nitrogen. Genomic DNA was extracted in four replicates using a buffer (100 mM Tris pH 8, 50 mM EDTA, 500 mM NaCl, 1.5% SDS), precipitated with isopropanol, washed with 70% ethanol, and dissolved in TE buffer. DNA purity was checked with NanoDrop and Qubit.

## Phenotypic analysis of *Arabidopsis thaliana rbp47abcc'*

A quadruple mutant of *RBP47* (*rbp47abcc'*) was obtained from Monika Chodasiewicz (KAUST, Thuwal, KSA). *rbp47abcc'* was generated in the Col-0 genetic background by clustered regularly interspaced short palindromic repeats (CRISPR)–Cas9 (Xie et al, 2023). Therefore, wild-type (Col-0) and *rbp47abcc'* mutant plants were germinated on ½ MS medium under standard growth conditions. Sterilized seeds were stratified at 4 °C for 2 days in the dark before being transferred to ½ MS agar plates (pH 5.8) for germination. After 5 days, seedlings of approximately equal length were carefully transferred to fresh ½ MS agar plates adjusted to three different pH conditions (5.8, 4.0, and 3.5) to assess growth under acidic stress. For the iron toxicity experiment, 10 mg/L FeNaEDTA was supplemented into the medium at pH 4.0 to create a high-iron condition (18 + 10 mg/L FeNaEDTA). All plates were incubated in a controlled growth chamber at 23 °C with a 16-h light/8-h dark photoperiod. After 15 days, plants were harvested, and their fresh weight was measured on day 16 to evaluate growth responses under different pH and iron conditions. All the experiments were conducted in biological triplicate.

### Ethics approval and consent to participate
Not applicable to this work.

## Microbiome analysis

### 16S sequencing
16S rDNA was amplified using V5-V7 primers (799F/1193R) with barcodes. PCR products were checked on agarose gel, purified, and sequencing libraries were prepared with NEB Next® Ultra DNA Library Prep Kit. Library quality was assessed with Qubit and Agilent Bioanalyzer and sequenced on an Illumina platform. Reads processing: Paired-end reads were merged using FLASH (V1.2.11, http://ccb.jhu.edu/software/FLASH/). Quality filtering was performed using the fastp (Version 0.23. 1) and the chimeric sequences were removed with the vsearch package (V2.16.0, https://github.com/torognes/vsearch).

### Denoise and annotation
Denoise was performed using the DADA2 module in the QIIME2 software (Version QIIME2-2022.2) to obtain ASVs (Amplicon Sequence Variants) and their annotations were obtained by mapping to the Silva and NCBI databases.

## Data normalization

The raw counts were transformed using the Hellinger method which involves two main steps: Relative Abundance Calculation and Square Root Transformation. To assess the diversity, richness, and evenness of the communities within the sample, alpha diversity was calculated using seven indices in QIIME2: Observed OTUs, Chao1, Shannon, Simpson, Dominance, Good's Coverage, and Pielou's Evenness and selected further for determining the Shannon index. To evaluate the complexity of community composition and compare differences between accessions, beta diversity was calculated based on Jaccard and Weighted Unifrac distances using the Phyloseq R package. Principal Coordinate Analysis (PCoA) was performed to get principal coordinates and visualize from complex and multidimensional data. The ordination score of PCOA1 axis was extracted holding the maximum variation factor and k-means clustering was performed and significance differences among the clusters were tested by ANOVA. The ordination scores were correlated with the relative abundances of seed-associated genera using the psych R package.

## Correlation network analysis

Microbial co-occurrence networks for the bacterial (genera) community having a significant correlation with PCOA1 of beta diversity were carried out with Spearman rank correlations. Only robust (Spearman's r > 0.8, or r < −0.8), and statistically significant ($p < 0.05$) correlations were considered. The visualization was done on Gephi (version 0.10.1) software (Jacomy et al, 2014).

## Geographical maps

The accession features were presented on the World map using R (version 4.5.1) packages rnaturalearth (version 1.1.0) and rnaturalearthdata (1.0.0) (https://cran.r-project.org/web/packages/rnaturalearth/vignettes/rnaturalearth.html).

## Distance-based analyses of β-diversity

To evaluate the influence of bacterial genera on seed microbiome β-diversity, we applied three complementary distance-based approaches using the R packages vegan (version 2.7.1) (Oksanen et al, 2025) and phyloseq (version 1.52.0) (McMurdie and Holmes, 2013). Weighted UniFrac distance matrices were generated from the phyloseq object (ASV abundance, taxonomy, and phylogeny). Envfit—genus-level abundances were fitted onto PCoA ordinations with 999 permutations to assess correlations with community structure. PERMANOVA (adonis2)—variance partitioning of β-diversity was tested against individual and combined genera using 999 permutations. Distance-based redundancy analysis (dbRDA/capscale) constrained ordination models were performed with genus abundances as predictors, and significance was evaluated via permutation-based ANOVA (999 permutations) distance-based analyses of β-diversity.

## Key taxa potential function prediction

Functional predictions were generated using PICRUSt2 (version 2.4.1) (Douglas et al, 2020; Langille et al, 2013), which infers gene family copy numbers per ASV based on reference genomes. We restricted our analysis to ASVs previously identified as belonging to the triad 1 and triad 2 groups (Dataset EV5). Within each triad, predicted KO copy numbers were summed and ranked, and the top 20 KOs were retained for downstream comparisons. Functional profiles were then compared to assess overlap and uniqueness between triad groups.

## Genome-wide association analysis

The genome-wide association study (GWAS) utilized single-nucleotide polymorphism (SNP) genotype data from the Bergelson laboratory (Horton et al, 2012) and Alonso-Blanco et al (2016; http://1001genomes.org/data/GMI-MPI/releases/v3.1/).

SNP markers underwent a stepwise filtering process to ensure data quality. Initially, variants with missing call rates exceeding 10% were excluded. Subsequently, accessions with missing call rates above 10% were also removed. Finally, SNP markers were retained only if they exhibited a minor allele frequency (MAF) of at least 5%. Following this filtering process, the final dataset comprised 875,991 biallelic markers and 215 samples, with an overall genotyping rate of 95.85%. All filtering steps were performed using PLINK (2.0) (Chang et al, 2015).

A linear mixed model GWAS was conducted using the EMMAX suite (version 2012021) (Kang et al, 2010). To account for cryptic relatedness among samples, the Balding-Nichols kinship matrix was computed using EMMAX-kin. Subsequently, EMMAX was used to perform association analyses with the phenotype. A genome-wide significance threshold of $P < 1 \times 10^{-5}$ was applied, as indicated by a red horizontal line in the Manhattan plot.

To identify genetic associations, the common KEGG Orthologs (KOs) found among the accessions with the two triads (either *Deltfia-Chrysobacterium-Stenotrophomanus* or *Paenibacillus-Cohnella-Brevibacillus*) were considered as target features. Their KO-predicted relative abundance values were obtained using PICRUSt2 and used as a trait. Further association analyses were performed with the relative abundance values of each triad.

# Data availability

16S sequencing data have been deposited in the NCBI SRA repository with the submission ID: SUB15365883 and project ID: PRJNA1273156 and the following URL link: https://dataview.ncbi.nlm.nih.gov/object/PRJNA1273156?reviewer=5ffe2peps1h6ljb72fprliq8ho.Genotype data is accessible at http://1001genomes.org/data/GMI-MPI/releases/v3.1/. Any additional data supporting the findings are available from the corresponding author upon reasonable request.

The source data of this paper are collected in the following database record: biostudies:S-SCDT-10_1038-S44319-025-00635-x.

# Peer review information

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

## Acknowledgements

The work was supported by KAUST grant BAS/01/1062-01-01 and RGC/3/5638-01-01 to HH. We want to thank Joy Bergelson for providing the SNP data and Monika Chodasiewicz for providing *rbp47abcc'* mutant seeds. We would also like to thank Santosh Satbhai and Andrea Gonzalez Munoz for their support in the initial stages of data analysis. We are also grateful to Arsheed Sheikh and the Hirt lab members for technical assistance and fruitful discussions.

## Author contributions

**Sabiha Parween**: Formal analysis; Methodology; Writing—original draft; Writing—review and editing. **Naheed Tabassum**: Formal analysis, Validation; Investigation; Writing—original draft. **Kirti Shekhawat**: Investigation; Writing—original draft. **Bruno Gnannt**: Investigation. **Waad Alzayed**: Investigation. **Rewaa Jalal**: Investigation. **Heribert Hirt**: Conceptualization; Supervision; Funding acquisition; Validation; Project administration; Writing—review and editing.

Source data underlying figure panels in this paper may have individual authorship assigned. Where available, figure panel/source data authorship is listed in the following database record: biostudies:S-SCDT-10_1038-S44319-025-00635-x.

## Disclosure and competing interests statement

The authors declare no competing interests.

# Expanded View Figures

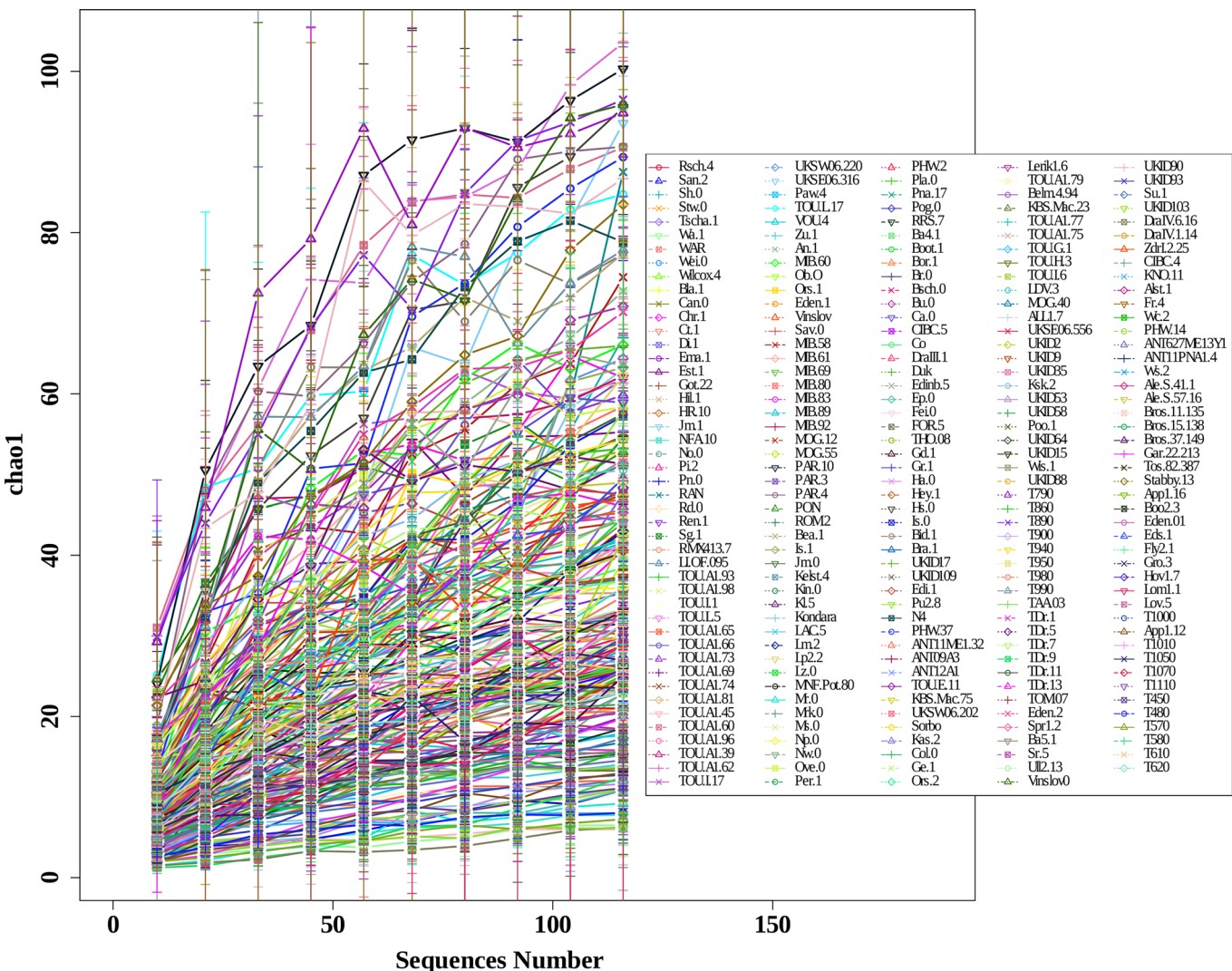

**Figure EV1. Rarefaction curve across the accessions.**

The richness measured by Chao1 across all the accessions is plotted over the sequence numbers. Each accession is highlighted by a different color in the plot. The curves show that species richness increases with sequencing effort, and several samples approach a plateau, suggesting that their microbial communities have been adequately sampled.

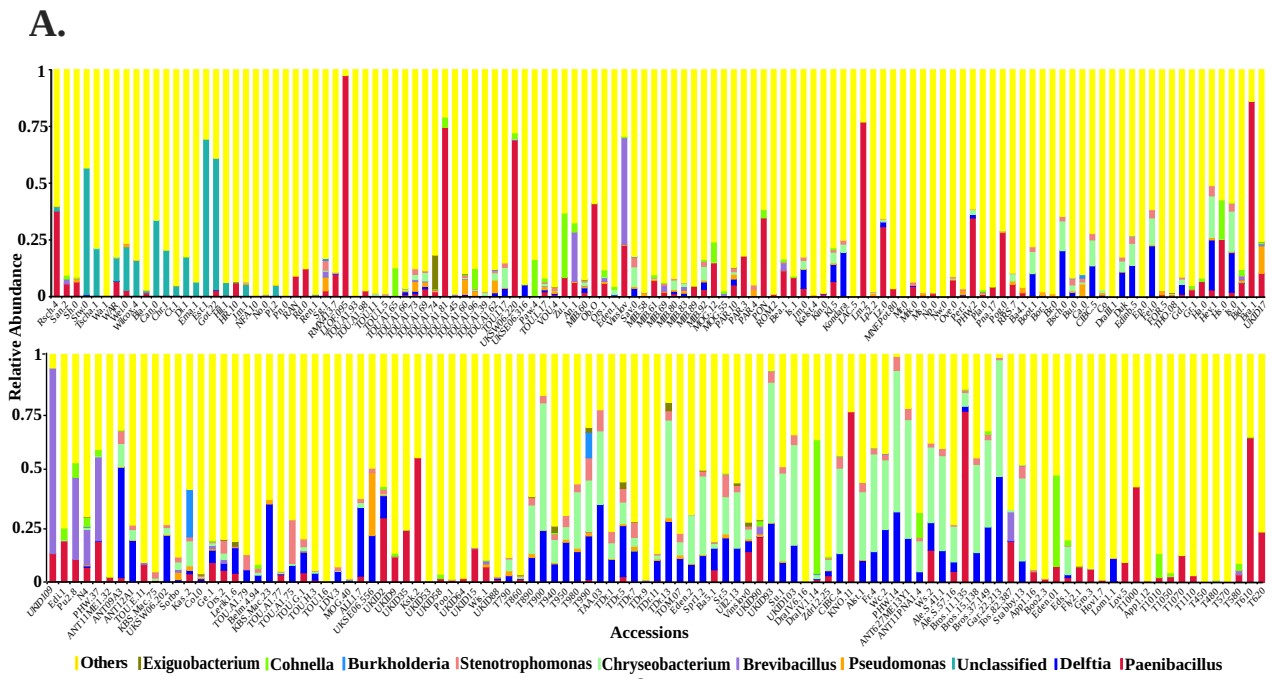

**A.**

**B.**

◀ **Figure EV2.** **(A)** Relative abundance of top 10 genera across all Arabidopsis accessions. **(B)** Presence of shared and unique genera of Arabidopsis seed microbiome across all accessions.

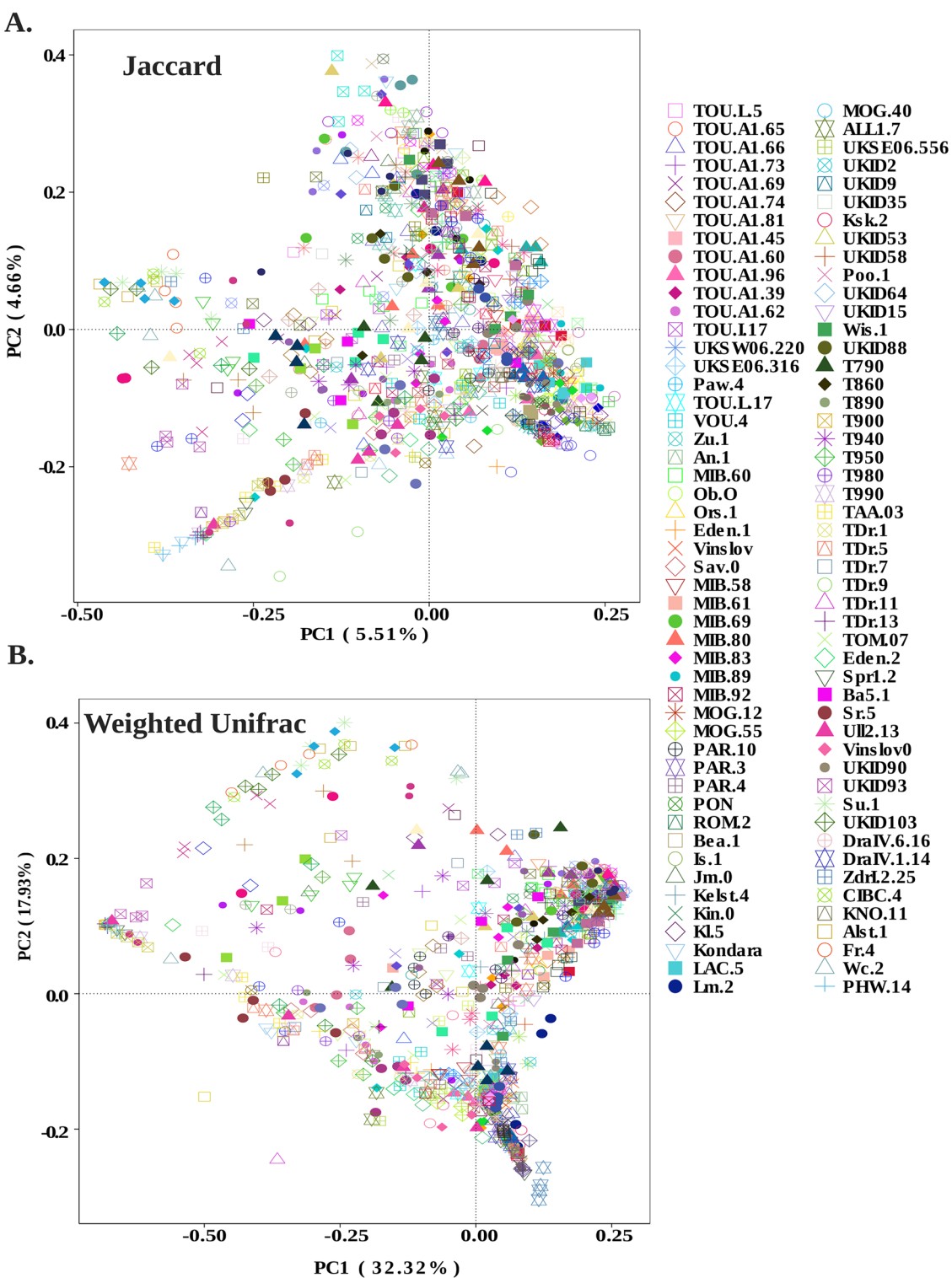

**Figure EV3. PCoA plot of beta diversity of Arabidopsis seed microbiome.**

Plots by Jaccard (A) and (B) Weighted Unifrac diversity measures.

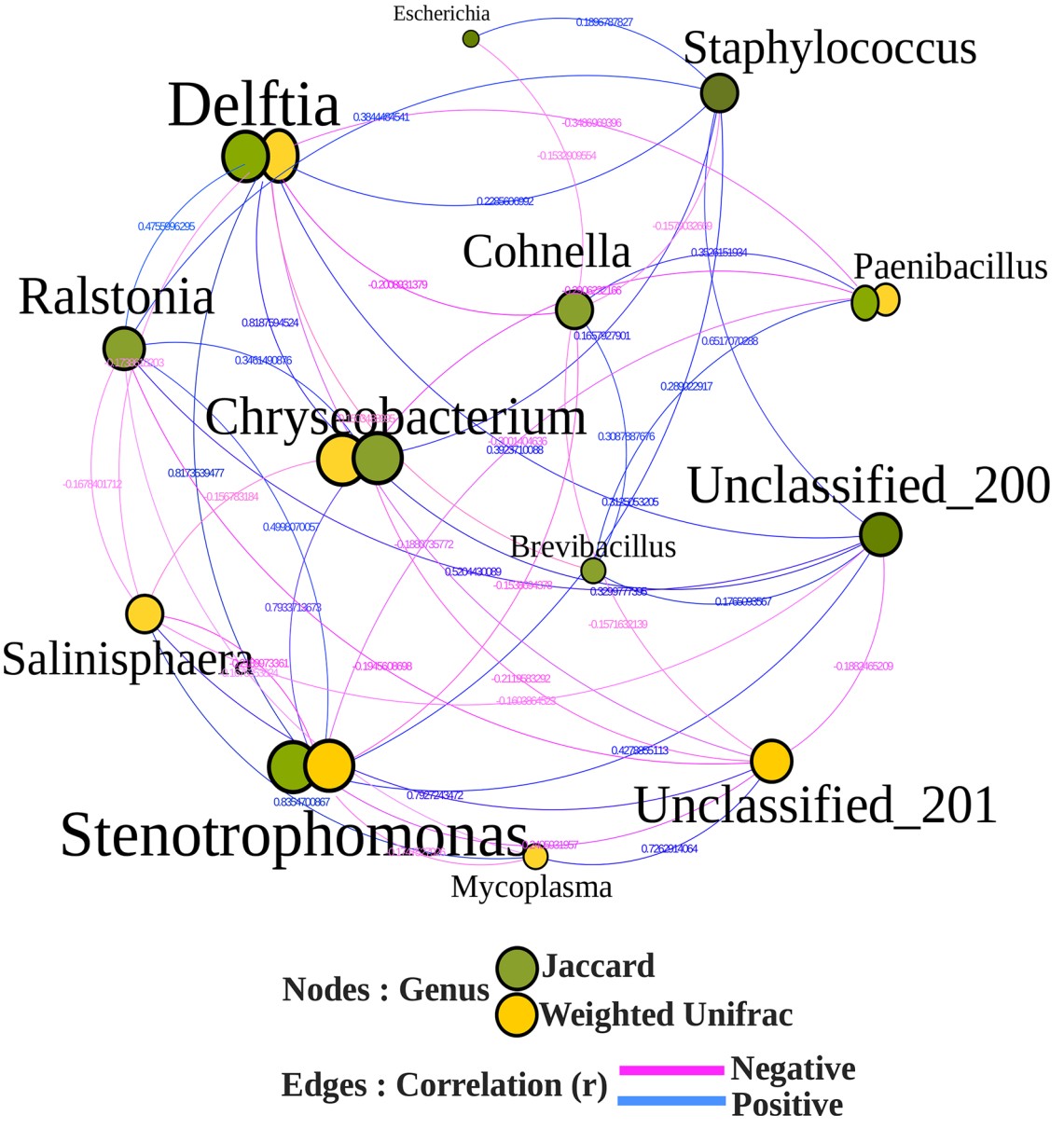

**Figure EV4. Network of genus abundance.**

The network shows the correlation patterns among genera in relation to Jaccard and Weighted UniFrac PC1 diversity measures. The color of the nodes represents their correlation occurrence with the diversity measures, while the node size corresponds to the degree of connectivity. Edges depict the Spearman correlation values, with positive correlations highlighted in blue and negative correlations in magenta.

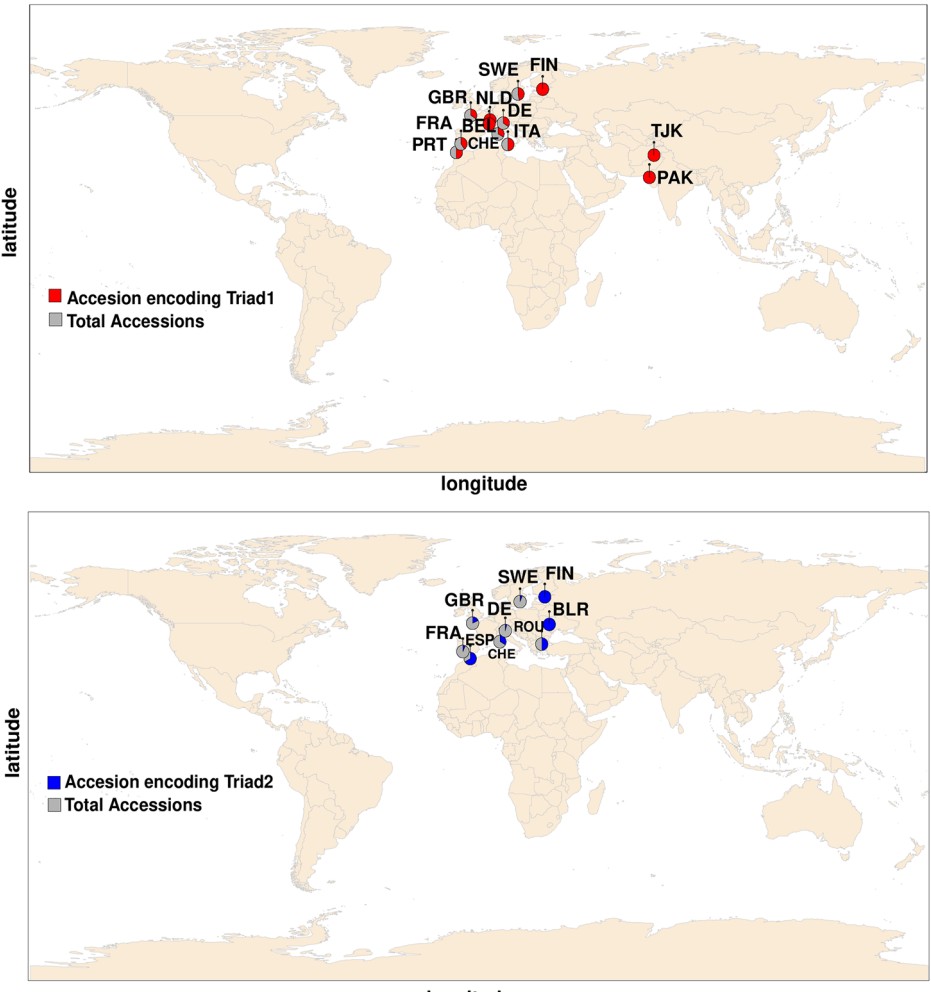

**Figure EV5. Accessions encoded by triads across different countries.**

Triad 1 includes *Chryseobacterium, Delftia, Stenotrophomonas* and Triad 2 *Paenibacillus, Brevibacillus and Cohnella*. We generated pie charts overlaid on a world map to represent each country's contribution to the total accessions versus those containing triads, defined by the presence across all genus of respective triad.

