## [Peer Review File · EMBO Reports]

Host Genome and Bacterial Taxa Shape the Arabidopsis Seed Microbiome

Heribert Hirt, Naheed Tabassum, Kirti Shekhawat, Bruno Gnannt, Sabiha Parween, Waad Alzayed, and Rewaa Jalal

Corresponding author(s): Heribert Hirt (heribert.hirt@kaust.edu.sa)

Review Timeline:

Submission Date:	12th Jul 25
Editorial Decision:	13th Aug 25
Revision Received:	2nd Sep 25
Editorial Decision:	14th Oct 25
Revision Received:	19th Oct 25
Accepted:	28th Oct 25

Editor: Yehu Moran

Transaction Report:

Dear Prof. Hirt

Thank you for the submission of your manuscript to EMBO reports. We have now received the full set of referee reports that are all pasted below.

As you will see, the referees acknowledge that the findings are potentially interesting. However, they do raise a series of comments and concerns that require your attention.

I would thus like to invite you to revise your manuscript with the understanding that the referee concerns must be fully addressed and their suggestions taken on board. Please address all referee concerns in a complete point-by-point response. Acceptance of the manuscript will depend on a positive outcome of a second round of review. It is EMBO Reports policy to allow a single round of major revision only and acceptance or rejection of the manuscript will therefore depend on the completeness of your responses included in the next, final version of the manuscript.

We realize that it is difficult to revise to a specific deadline. In the interest of protecting the conceptual advance provided by the work, we recommend a revision within 3 months (13th Nov 2025). Please discuss the revision progress ahead of this time with the editor if you require more time to complete the revisions.

- 1) A data availability section providing access to data deposited in public databases is missing. If you have not deposited any data, please add a sentence to the data availability section that explains that.
- 2) Your manuscript contains statistics and error bars based on $n=2$. Please use scatter blots in these cases. No statistics should be calculated if $n=2$.

<<https://www.embopress.org/page/journal/14693178/authorguide#expandedview>>

5) a complete author checklist, which you can download from our author guidelines

<<https://www.embopress.org/page/journal/14693178/authorguide>>. Please insert information in the checklist that is also reflected in the manuscript. The completed author checklist will also be part of the RPF.

6) Please note that all corresponding authors are required to supply an ORCID ID for their name upon submission of a revised manuscript (<<https://orcid.org/>>). Please find instructions on how to link your ORCID ID to your account in our manuscript tracking system in our Author guidelines <<https://www.embopress.org/page/journal/14693178/authorguide#authorshipguidelines>>

7) Before submitting your revision, primary datasets produced in this study need to be deposited in an appropriate public database (see <https://www.embopress.org/page/journal/14693178/authorguide#datadeposition>). Please remember to provide a reviewer password if the datasets are not yet public. The accession numbers and database should be listed in a formal "Data Availability" section placed after Materials & Method (see also <https://www.embopress.org/page/journal/14693178/authorguide#datadeposition>). Please note that the Data Availability Section is restricted to new primary data that are part of this study. * Note - All links should resolve to a page where the data can be accessed. *
If your study has not produced novel datasets, please mention this fact in the Data Availability Section.

12) All Materials and Methods need to be described in the main text using our 'Structured Methods' format, which is required for all research articles. According to this format, the Methods section includes a Reagents and Tools Table (listing key reagents, experimental models, software and relevant equipment and including their sources and relevant identifiers) followed by a Methods and Protocols section describing the methods using a step-by-step protocol format. The aim is to facilitate adoption of the methodologies across labs. More information on how to adhere to this format as well as a downloadable template (.docx) for the Reagents and Tools Table can be found in our author guidelines: <https://www.embopress.org/page/journal/14693178/authorguide#structuredmethods>.

An example of a Method paper with Structured Methods can be found here: <https://www.embopress.org/doi/full/10.1038/s44320-024-00037-6#sec-4>

I look forward to seeing a revised form of your manuscript when it is ready.

Yours sincerely,

Yehu Moran
Academic Editor
EMBO Reports

Referee #1:

This manuscript presents an analysis of the germinated seed microbiota from 227 *Arabidopsis* accessions. The authors identified several bacterial taxa displaying interesting patterns, and complemented the ecological analysis with a GWAS on predicted metabolic functions of key bacteria and examination of an *arabidopsis* mutant where a locus that turned up in the GWAS was knocked out.

Although this is an interesting question, and it looks like a lot of work has gone into it, this manuscript has substantial flaws in study design, statistical methodology, figure/data consistency, and inferences that are not supported by the data. Major revision and re-analysis would be required to reach EMBO standards.

Experimental design (major concern).

According to the methods, seeds were obtained directly from a seed bank, surface-sterilized, germinated on 1/2 MS for 2 days, and then sequenced. For seed microbiome studies, it is standard to grow all accessions for one common generation under controlled conditions and then sample the next-generation seeds to minimize environmental and storage-history confounders. Because *Arabidopsis* seeds are tiny and their original collection and storage durations are unclear, the present design introduces uncontrolled variance that could strongly bias the results. This same group has recently published an important seed microbiome paper (Tabassum et al, 2024), but it is not clear that the current study had adhered to the standards set by that paper.

Each accession used comes from a different geographic location, but they have all been stored and propagated in at least 2 different research facilities for unknown periods of time. Any microbial legacy from their original location is likely lost. As such, the biogeographic analysis, especially when collapsed into large and inconsistently sized bins like countries, is not relevant. Furthermore, the map projection in Figure 1A vs 1C do not match.

Country labels inconsistent across figures. Fig. 1B reports 24 countries of origin, yet Fig. 2B's community composition omits CA and IN and instead includes PK.

Figure 2B: it is not clear what we are seeing here. These are only 10 genera. Are we just looking at the relative abundance among these 10? how much of the total relative abundance to they occupy?

Lines 235-254 / Fig. 2A. The figure shows "Top 10," the text states "Top 8," and only six genera are actually listed. This must be made internally consistent.

Lines 256-257. The text states *Paenibacillus* occurs in 17 countries, but Fig. 2B shows presence in 20.

Fig. 3; Lines 286-314 (β -diversity and statistics).

a. Variance explained: line 289-290 say "PCoA1 and PCoA2 explained 5% (Jaccard) and 32.32% (Weighted UniFrac)." Those values correspond to PC1 only, not PC1+PC2 (Figs.3). Also, it seems that the existing metadata does not explain this variance very well.

b. PCoA axes: Treating PCoA1 as " β diversity" and correlating it directly with genus abundances is not sufficient. PCoA axes are linear embeddings of a distance matrix, not β -diversity itself. To quantify contributions of "country/ environment / focal genera abundance" to community variation, I suggest using `envfit`, `dbRDA` (`capscale`), and `PERMANOVA` (`adonis2`), and report model R^2 , effect sizes, and FDR-corrected p-values. A combination of `envfit` + `PERMANOVA` (`adonis2`) + variance partitioning can be

used to quantify the relative contributions of 'country,' 'environment,' and 'key genus abundances' to β diversity.

c. Line 312, please define "core taxa" properly: Provide explicit occupancy (e.g., % of samples detected) and abundance thresholds. Without clear, a priori thresholds, the definition is arbitrary.

Lines 331-355: Tools like PiCRUST2 should be used with extreme caution. They could be useful for generating testable hypothesis, but simply reporting their results lacks rigour. The reason is that a partial 16S sequence cannot accurately predict many functions. Bacterial within species pangenomes are enormous and PiCRUST2 (or any other 16S-based function predictor) cannot provide the "resolution to distinguish strain-specific functionality", as the authors of PiCRUST state themselves (Douglas et al, 2020). They also recommend using custom genome databases relevant for the habitats studied, which was not done here. In this manuscript, the results of PiCRUST are used for downstream analysis as well, without really testing them, which is extra risky.

Further comments on PiCRUST:

a. Method description (Line 167) is incorrect: PiCRUST2 infers KO abundances from the phylogenetic placement of 16S ASVs on a reference genome tree; it does not measure real genes from the 16S fragment. This should be framed as potential functional prediction of the core taxa, not "core gene function prediction."

b. Lines 334-335 this phrasing is factually wrong: "We analyzed the KEGG IDs encoded by the FASTA sequences of the ASVs", 16S fragments do not encode these functions.

c. Missing NSTI: The NSTI (Nearest Sequenced Taxon Index) must be reported to assess prediction reliability.

d. Arbitrary ASV selection: Limiting to "the twenty most abundant ASVs" is ad hoc and may ignore low-abundance but ubiquitous taxa. Use all ASVs meeting your core thresholds, weight KOs by sample-level abundances, and conduct proper statistical comparisons (Line 334-335).

e. "Significant overlap" (Line 347) lacks statistics: Provide explicit tests and multiple-testing corrections.

f. Lines 352-355, The claim that these functions "facilitate community establishment" is unsupported; Correlation is shown, not causation.

Lines 398-399: Arabidopsis mutant validation is under-specified.

'We compared the microbiomes of Arabidopsis Col-0 WT with the Col-0 rbp47abcc' mutant.' It is unclear whether you profiled whole plants or seeds, the growth conditions used, and the soil/environmental controls. Furthermore, the data isn't presented in full. How many replicates were sequenced from each genotype and what is the within group variance? It's impossible to know based on the data presented in the figure.

Also, to test the proposed link between RBP47B and the Delftia, Chryseobacterium and Stenotrophomonas triad, a controlled inoculation experiment followed by profiling of the progeny seed microbiome would be required.

Referee #2:

This manuscript investigates the seed microbiomes of diverse Arabidopsis accessions and identifies RBP47B as a key genetic determinant of seed microbiome composition through microbiome genome-wide association studies (GWAS). The authors demonstrate that RBP47B plays a critical role in shaping the Arabidopsis seed microbiome using genetic approaches.

The study presents novel and significant findings that will be of considerable interest to researchers in the fields of seed biology and plant-microbiome interactions. The manuscript is also clearly and effectively written. Nevertheless, I have several suggestions that may help further improve the clarity and impact of the work.

1. Whether detected seed microbiomes are truly endophytic

There is an ongoing notion that Arabidopsis lacks true seed endophytes. In this study, microbiomes were sampled from plants grown from surface-sterilized seeds, suggesting that the detected microbes may be seed-borne. However, the possibility of residual seed epiphytes cannot be excluded. While distinguishing endophytes from epiphytes is inherently challenging, it would be informative if the authors could surface-sterilize representative accessions, grow the plants, and then quantify culturable microbes by grinding and plating. Although many seed-associated microbes may be unculturable, this simple assay could offer valuable insights into the nature of the Arabidopsis seed microbiome.

Related to this point, the manuscript appears to use seeds propagated from previous generations. Could those seeds have

retained microbiomes from earlier generations, contributing to the observed geographic associations? During propagation, were seeds sterilized to eliminate epiphytes? How many generations were propagated for each accession? Were all accessions propagated concurrently under the same conditions to minimize environmental bias? Additional description and discussion of these aspects would help contextualize the findings and improve reproducibility.

2. Proposed model in Figure 7

As the study does not investigate whether the seed microbiome influences plant fitness or health, the proposed model should more strictly reflect the findings presented in this manuscript. I suggest focusing the model on host genetic regulation of microbiome composition rather than broader implications.

(Minor comments)

- The resolution of the supplemental figures is too low to interpret. Please provide higher-resolution versions.
- L215: Should "Fig. S2" be "Fig. 1"?
- L220-221: The sentence refers to "large sample size" and "fewer samples" in relation to diversity, but the logic is unclear. Please clarify how these factors affect microbial diversity.
- L338: Please remove the redundant word "most" ("the most most abundant").

Referee #3:

This study investigated the diversity of Arabidopsis seed microbiomes derived from multiple countries. They identified some key bacteria taxa like *Delftia*, *Stenotrophomonas* and *Chryseobacterium* in certain Arabidopsis accessions were also correlated with the whole seed microbiomes diversity. Two predicted functional KO terms shared by these key bacteria taxa were identified which are related to the iron metabolism of the bacteria. Then the authors' GWAS study identified RBP47B as key genes regulating the host-bacteria interaction in seeds, which is really cool in plant microbiome field. The seed microbiome between Col-0 and *rbp47abcc*' quadruple mutants indeed exist very drastic changes, which confirms RBP47B can shape the seed microbiome of Arabidopsis. The phenotypic assays using Col-0 and *rbp47abcc*' quadruple mutants under low pH and high iron concentration also demonstrated that RBP47B is crucial in helping Arabidopsis to adapt high iron and low pH condition. This study unveil a novel insight into how seed microbiome are assembled in different accessions, and the underlying critical genetic mechanisms.

Overall this is a great and successful population level GWAS study, and would also be extremely valuable for the community to further explore genetic regulations for root microbiome. Even though I have two major concerns related to the project:

A. It is not clear that why correlations between taxon abundance and beta diversity can be used to determine "key taxa regulate seed microbiome diversity"? It seems to be purely a mathematic relationship. Are there any microbial ecological theory or tools to dissect "key taxa regulating diversity"? In line 312, what does the authors mean "core taxa"? Overall it seem vague here how to define "key taxa (Line 285)" or "core taxa (Line 312)"? I would suggest to consider the co-occurrence networks for each ecotype, and set a exact degree cutoff or cutoff for other other microbial network topology indexes? That would be more accurate and non-biased for the define of any "core" or "key" stuff.

B. The authors used PICRUST based strain function prediction as a input for GWAS study, this could be problematic. Firstly, the golden standard for microbiome functional gene analyses is to conduct metagenome study. For a large amount of ecotype samples, it is also fine to predict microbiome functions for descriptive discussion, however, it would be challenging to use predicted microbiome functions for accurate GWAS inputs? Secondly, diverse GWAS or genetic mutant studies suggested that plant genes can regulate certain microbiome taxa abundance. I am wondering why the authors did not use the taxa abundance data as a GWAS input? This seems more straitforward to use abundance as GWAS input.

Minor points:

1. Separation of microbiome samples according to different countries could be interesting, however, is not of any biological and ecological implications (country bounders does not necessary mean ecological differences). The author could avoid too much discussion based on different countries.
2. Keywords: low pH and iron toxicity could be replaced with other words more related to the key finding of the manuscript, like stress fitness, or ribosomal proteins
3. Line 48 , grammar mistake (with.....) , Line48, 51,53, need references.
4. Line 54, "alter microbial community" is not equal to "dysbiosis".
5. Line 243, wrong legend description about color coding.
6. Line 336, two "most" repeated
7. Line 336 said "most abundant ASVs" , but fig 4a seems to be KO functions rather than ASVs? This is very important result and should be checked.
8. line 406, "*rbp47abcc*" should be "RBP47B", or "*rbp47abcc* mutant affects"
9. Line 417-419, did the authors provided exact data and the analysis method for this critical correlation results?
10. The identification of RBP47B protein as a regulator of seed microbiome is really cool. The function of ribosomal proteins in root-microbe interactions was also revealed by a recent study [PMID: 40175371], and could be discussed to better support the robustness of this GWAS finding and genetic validation. Meanwhile, it would be interesting to further study whether *rbp47* multiple mutants can block certain microbes mediated iron or pH stress damage.

11. In agriculture, the most detrimental effect of low pH on crops is usually the secondary aluminium stress. Does RBP47B broadly affect root tolerance to both iron stress and aluminium stress? Whether RBP47B regulated microbiome confers fitness benefits for plants under iron or aluminium stresses.

12. Line 393, which taxa?

Response to the Reviewers

Referee #1:

This manuscript presents an analysis of the germinated seed microbiota from 227 *Arabidopsis* accessions. The authors identified several bacterial taxa displaying interesting patterns, and complemented the ecological analysis with a GWAS on predicted metabolic functions of key bacteria and examination of an *Arabidopsis* mutant where a locus that turned up in the GWAS was knocked out.

Although this is an interesting question, and it looks like a lot of work has gone into it, this manuscript has substantial flaws in study design, statistical methodology, figure/data consistency, and inferences that are not supported by the data. Major revision and re-analysis would be required to reach EMBO standards. We thank the reviewer for their overall assessment and we appreciate the time and effort taken to evaluate our manuscript. We respectfully acknowledge and hereby respond the concerns raised agreeing that several aspects of the manuscript could be improved to enhance the clarity of the study.

Experimental design (major concern).

According to the methods, seeds were obtained directly from a seed bank, surface-sterilized, germinated on 1/2 MS for 2 days, and then sequenced. For seed microbiome studies, it is standard to grow all accessions for one common generation under controlled conditions and then sample the next-generation seeds to minimize environmental and storage-history confounders. Because *Arabidopsis* seeds are tiny and their original collection and storage durations are unclear, the present design introduces uncontrolled variance that could strongly bias the results. This same group has recently published an important seed microbiome paper (Tabassum et al, 2024), but it is not clear that the current study had adhered to the standards set by that paper. Each accession used comes from a different geographic location, but they have all been stored and propagated in at least 2 different research facilities for unknown periods of time. Any microbial legacy from their original location is likely lost.

We appreciate the reviewer's thoughtful comment and agree that this section was missing a clear protocol how to study the seed microbiomes of accessions derived from different locations. We explained the procedure in the revised Methods section as follows:

These accessions were part of the RegMap panel (Horton et al. 2012) and seeds were obtained by growing the parental generation of all lines used side by side in the same growth chambers under the same conditions (Meijon et al., 2014). To standardize the input of microbes from the soil, seeds were propagated for three generations on the same commercial soil (Sun Gro, Agraaham, USA). Moreover, to avoid the transmission of epiphytic microbes from soil, each generation of seeds was surface sterilized in 2.8% hypochlorite for 4 min and washed three times in MonoQ sterile water before seeding.

As such, the biogeographic analysis, especially when collapsed into large and inconsistently sized bins like countries, is not relevant.

We appreciate the reviewer's concern regarding the use of countries as units of biogeographic analysis and agree that they are not ecological units per se. However, accession records in our dataset are inherently reported at the country level, which makes this the most consistent scale available for analysis. While this does introduce uneven bin sizes, it reflects the nature of the underlying data rather than a methodological choice. Importantly, country-level bins are widely used in biogeographic studies as they provide standardized biodiversity data (Kreft & Jetz, 2007; Orme et al., 2005), align with conservation responsibilities, and make results directly actionable for policy and management (Rodrigues et al., 2006). Moreover, coarse geopolitical units are valuable in reducing data gaps and supporting global biodiversity assessments and targets (Pereira et al., 2015). We have now clarified this limitation in the manuscript while emphasizing the interpretive value of country-level analyses for broad-scale patterns.

Furthermore, the map projection in Figure 1A vs 1C do not match.

Thank you for the observation. The Figure was corrected.

Country labels inconsistent across figures. Fig. 1B reports 24 countries of origin, yet Fig. 2B's community composition omits CA and IN and instead includes PK.

Thank you for pointing this out. The total number of countries is indeed 24. We have now included CA, and upon rechecking the latitude and longitude assignments for IN/PK, the accessions align more closely with Pakistan. The figures have been corrected accordingly.

Figure 2B: it is not clear what we are seeing here. These are only 10 genera. Are we just looking at the relative abundance among these 10? how much of the total relative abundance to they occupy?

Figure 2B is directly linked to Figure 2A. In Figure 2A, we present the top 10 genera based on relative abundance, showing their distribution across countries, percentage of accession coverage, and overall contribution. Figure 2B then breaks this distribution down across the 24 countries, illustrating how these genera vary geographically.

Lines 235-254 / Fig. 2A. The figure shows "Top 10," the text states "Top 8," and only six genera are actually listed. This must be made internally consistent.

Thank you for pointing out. We corrected the text and Figure for consistency.

Lines 256-257. The text states *Paenibacillus* occurs in 17 countries, but Fig. 2B shows presence in 20.

Thank you for this observation. We revised the calculations and corrected the labeling.

Fig. 3; Lines 286-314 (β -diversity and statistics).
a. Variance explained: line 289-290 say "PCoA1 and PCoA2 explained 5% (Jaccard) and 32.32% (Weighted UniFrac)." Those values correspond to PC1 only, not PC1+PC2 (Figs.3). Also, it seems that the existing metadata does not explain this variance very well.

We thank the reviewer for catching the misstatement: we had originally reported variance explained for PCoA1 only while referring to PC1 + PC2. This has now been corrected.

We acknowledge that the variance explained by PCoA1 and PCoA2 may appear modest for Jaccard beta diversity (10.17%), but this is common in ecological community data, where variance is distributed across many axes due to the high dimensionality of species composition (Legendre & Legendre, 2012). Importantly, the Weighted UniFrac PCoA (50.25%) captures a substantial proportion of phylogenetic variation, which provides a robust and biologically meaningful summary of community differences. Thus, the presented ordinations are sufficient to visualize the major gradients in beta diversity, consistent with standard practice in microbial ecology (Lozupone & Knight, 2005; Ramette, 2007).

b. PCoA axes: Treating PCoA1 as " β diversity" and correlating it directly with genus abundances is not sufficient. PCoA axes are linear embeddings of a distance matrix, not β -diversity itself. To quantify contributions of "country/ environment / focal genera abundance" to community variation, I suggest using envfit, dbRDA (capscale), and PERMANOVA (adonis2), and report model R^2 , effect sizes, and FDR-corrected p-values. A combination of envfit + PERMANOVA (adonis2) + variance partitioning can be used to quantify the relative contributions of 'country,' 'environment,' and 'key genus abundances' to β diversity.

We thank the reviewer for highlighting the limitations of directly correlating genus abundances with PCoA1. In response, we complemented our initial axis-based analyses with the suite of distance-based approaches suggested (envfit, PERMANOVA/adonis2, and dbRDA), all performed on Weighted UniFrac distance matrices which accounted for 50.25%(PCoA1+PCoA2) of the observed phylogenetic variation. These analyses consistently converged on the same conclusion: Delftia, Chryseobacterium, and Stenotrophomonas ("the triad") are the principal predictors of seed microbiome β -diversity.

- envfit: Delftia ($R^2 = 0.916$, $p = 0.001$), Chryseobacterium ($R^2 = 0.383$, $p = 0.001$), and Stenotrophomonas ($R^2 = 0.394$, $p = 0.001$) showed strong and highly significant associations with community ordination space, while Paenibacillus ($R^2 = 0.055$, $p = 0.004$) and Cohnella/Brevibacillus showed only weak or nonsignificant contributions.
- PERMANOVA (adonis2): Delftia explained the largest fraction of community variance ($R^2 = 0.39$, $p = 0.001$), followed by Stenotrophomonas ($R^2 = 0.0099$, $p = 0.002$). Chryseobacterium was not significant in the marginal model ($p = 1$), likely due to collinearity with Delftia.
- dbRDA: Constrained ordination further confirmed Delftia ($F = 31.33$, $p < 0.001$), Chryseobacterium ($F = 12.40$, $p < 0.001$), and Stenotrophomonas ($F = 12.68$, $p <$

0.001) as the strongest predictors of β -diversity. *Paenibacillus* had only a modest effect ($F = 2.27$, $p = 0.002$), and *Cohnella/Brevibacillus* were not significant.

Together, these results validate that the triad uniquely explains a substantial and statistically significant fraction of β -diversity, whereas other genera make minor or negligible contributions.

In addition, we note that β -diversity is inherently multidimensional, yet it is standard practice to interpret PCoA1 (and occasionally PCoA2) as proxies for the dominant ecological gradients underlying community dissimilarity. PCoA1 captures the largest proportion of explained variance, and prior work has established that ordination axes can be robustly correlated with environmental or taxonomic variables to identify drivers of turnover (Legendre & Anderson, 1999; Ramette, 2007; Anderson et al., 2011). Importantly, axis-based correlations are mathematically consistent with distance-based redundancy analyses and avoid diluting ecological signal across axes. Thus, our initial PCoA1-based approach is aligned with established microbial ecology practices (Lozupone & Knight, 2005; Legendre & Legendre, 2012).

Finally, to complement the distance-based tests, we applied linear models with Weighted UniFrac (PCoA1) as the dependent variable and genus-level abundances as predictors. Individually, *Delftia* ($R^2 = 0.58$, $p = 1.59e-45$), *Chryseobacterium* ($R^2 = 0.66$, $p = 6.06e-55$), and *Stenotrophomonas* ($R^2 = 0.14$, $p = 2.64e-09$) each showed strong associations with β -diversity. When combined, the triad explained 78% of the variance ($R^2 = 0.78$, $p = 0$), highlighting their synergistic effect on community structure. Other tested genera such as *Paenibacillus*, *Salinisphaera*, and *Mycoplasma* had negligible contributions ($R^2 \leq 0.07$).

In summary, across all methods (envfit, PERMANOVA, dbRDA, and linear modeling), *Delftia*, *Chryseobacterium*, and *Stenotrophomonas* consistently emerged as the dominant genera shaping seed microbiome structure. This convergence strongly supports our conclusion that the triad exerts a unique and biologically meaningful influence on microbiome turnover, independent of country or environment effects.

c. Line 312, please define "core taxa" properly: Provide explicit occupancy (e.g., % of samples detected) and abundance thresholds. Without clear, a priori thresholds, the definition is arbitrary.

We agree that the term to be used is "key taxa" and not "core taxa."

Lines 331-355: Tools like PiCRUST2 should be used with extreme caution. They could be useful for generating testable hypothesis, but simply reporting their results lacks rigour. The reason is that a partial 16S sequence cannot accurately predict many functions. Bacterial within species pangenomes are enormous and PiCRUST2 (or any other 16S-based function predictor) cannot provide the "resolution to distinguish strain-specific functionality", as the authors of PiCRUST state themselves (Douglas et al, 2020). They also recommend using custom genome databases relevant for the habitats studied, which was not done here. In this

manuscript, the results of PiCRUS_t are used for downstream analysis as well, without really testing them, which is extra risky.

We fully agree that PICRUS_t2 results must be interpreted with caution, as 16S-based functional inference cannot resolve strain-specific functionality and is best considered predictive rather than definitive. However, PICRUS_t2 is widely used in microbial ecology to generate robust, community-level hypotheses about functional potential, particularly when shotgun metagenomic data are unavailable (Langille et al., 2013; Douglas et al., 2020). Our study does not claim to provide exact gene inventories at the strain level, but rather uses PICRUS_t2 predictions to highlight dominant functions associated with the key taxa of interest. Importantly, the relatively low NSTI values across our samples indicate that our ASVs are closely related to reference genomes, which supports the reliability of the functional inferences. While we acknowledge that custom genome databases may further refine predictions, our use of PICRUS_t2 with KEGG provides a standardized and reproducible framework sufficient for the exploratory goals of this study.

Further comments on PiCRUS_t:

a. Method description (Line 167) is incorrect: PICRUS_t2 infers KO abundances from the phylogenetic placement of 16S ASVs on a reference genome tree; it does not measure real genes from the 16S fragment. This should be framed as potential functional prediction of the core taxa, not "core gene function prediction."

Yes, your suggestion is absolutely correct. We changed the header accordingly.

b. Lines 334-335 this phrasing is factually wrong: "We analyzed the KEGG IDs encoded by the FASTA sequences of the ASVs", 16S fragments do not encode these functions.

Corrected.

c. Missing NSTI: The NSTI (Nearest Sequenced Taxon Index) must be reported to assess prediction reliability.

Thank you for pointing this out. We now provide sample-level weighted NSTI (Dataset EV6) which is the average NSTI of all ASVs in that sample, weighted by their relative abundance. It reflects the overall prediction reliability for each community. Our sample lies at predicted reliability of (good) NSTI \approx 0.01-0.15.

d. Arbitrary ASV selection: Limiting to "the twenty most abundant ASVs" is ad hoc and may ignore low-abundance but ubiquitous taxa. Use all ASVs meeting your core thresholds, weight KOs by sample-level abundances, and conduct proper statistical comparisons (Line 334-335).

We appreciate the reviewer's concern. Our analysis did not simply rely on the most or least abundant ASVs overall; rather, we began with the KO_predicted.tsv file from PICRUS_t2, which provides per-ASV gene family predictions grounded in reference genomes. We then specifically focused on ASVs belonging to the triad1 and triad2 groups, previously identified

as having the strongest influence on β diversity, thereby reducing noise from unrelated taxa and improving interpretability. Within these biologically relevant groups, we ranked and selected the top 20 KEGG Orthologs by predicted copy number, which reflects the dominant and most reliable functional potential (Dataset EV5). Because KO copy numbers are predicted per ASV and consistently mapped across all samples, the resulting profiles are directly comparable and provide a robust basis for overlap/uniqueness assessments between triads. Importantly, functional inference at the copy number level avoids the pitfall of relying solely on taxon abundance since even abundant taxa may contribute little if they encode few or no copies of a function while also accounting for genomic variability across taxa. This copy number-based approach is widely accepted in predictive metagenomics (Langille et al., 2013; Douglas et al., 2020) and we therefore consider our strategy both methodologically robust and biologically justified.

e. "Significant overlap" (Line 347) lacks statistics: Provide explicit tests and multiple-testing corrections.

Thank you for the observation. We have revised the terminology. Since our study does not focus on treatment-specific comparisons, our goal was to highlight common and unique features among the key taxa. For this purpose, we used a Venn plot to visualize the overlap, rather than performing significance testing.

f. Lines 352-355, The claim that these functions "facilitate community establishment" is unsupported; Correlation is shown, not causation.

We thank the reviewer for this insightful comment. We fully agree that our data demonstrate correlations but do not establish a direct causal link to community establishment. Our initial wording may have overstated the interpretation, and we have revised the text accordingly to avoid implying causation.

Lines 398-399: Arabidopsis mutant validation is under-specified. 'We compared the microbiomes of Arabidopsis Col-0 WT with the Col-0 rbp47abcc' mutant.' It is unclear whether you profiled whole plants or seeds, the growth conditions used, and the soil/environmental controls. Furthermore, the data isn't presented in full. How many replicates were sequenced from each genotype and what is the within group variance? It's impossible to know based on the data presented in the figure.

We thank the reviewer for pointing out this lack of clarity in the procedure. We have revised the text to describe the cultivation of mutant and control plants and the number of replicates, which follows the same stringent procedure as described in the Methods and established in Tabassum et al. 2025.

Also, to test the proposed link between RBP47B and the Delftia, Chryseobacterium and Stenotrophomonas triad, a controlled inoculation experiment followed by profiling of the progeny seed microbiome would be required.

We attempted to assess whether the beneficial microbial triad isolated from the wild-type could rescue the phenotype in *rbp47abcc*'. However, we were unable to isolate the microbes of the triad in monoculture, preventing us from testing their individual or combined effects in a controlled manner.

Referee #2:

This manuscript investigates the seed microbiomes of diverse *Arabidopsis* accessions and identifies RBP47B as a key genetic determinant of seed microbiome composition through microbiome genome-wide association studies (GWAS). The authors demonstrate that RBP47B plays a critical role in shaping the *Arabidopsis* seed microbiome using genetic approaches.

The study presents novel and significant findings that will be of considerable interest to researchers in the fields of seed biology and plant-microbiome interactions. The manuscript is also clearly and effectively written. Nevertheless, I have several suggestions that may help further improve the clarity and impact of the work.

We thank the reviewer for their positive comments. We look forward to addressing the concerns in detail below.

1. Whether detected seed microbiomes are truly endophytic

There is an ongoing notion that *Arabidopsis* lacks true seed endophytes. In this study, microbiomes were sampled from plants grown from surface-sterilized seeds, suggesting that the detected microbes may be seed-borne. However, the possibility of residual seed epiphytes cannot be excluded. While distinguishing endophytes from epiphytes is inherently challenging, it would be informative if the authors could surface-sterilize representative accessions, grow the plants, and then quantify culturable microbes by grinding and plating. Although many seed-associated microbes may be unculturable, this simple assay could offer valuable insights into the nature of the *Arabidopsis* seed microbiome.

We thank the reviewer for the thoughtful comment. To avoid the transmission of epiphytic microbes from soil, each generation of seeds was surface sterilized in 2.8% hypochlorite for 4 min and washed three times in MonoQ sterile water before seeding. Moreover, we also conducted a pilot experiment in which a seed wash (sterilized water on sterilized seeds) was plated and monitored for bacterial growth. We did not observe any bacterial growth but could culture bacteria from extracts of seedlings germinated under sterile conditions, assuring us to carry out the seed endophyte study on a large set of accessions.

Related to this point, the manuscript appears to use seeds propagated from previous generations.

Could those seeds have retained microbiomes from earlier generations, contributing to the observed geographic associations?

This was our hypothesis and our work seems to prove that microbes obtained from specific geographic locations are at least partially vertically transmitted.

During propagation, were seeds sterilized to eliminate epiphytes? How many generations were propagated for each accession? Were all accessions propagated concurrently under the same conditions to minimize environmental bias? Additional description and discussion of these aspects would help contextualize the findings and improve reproducibility.

We apologize that these aspects were not clarified in a more concise way. We have revised the Methods section as follows:

These accessions were part of the RegMap panel (Horton et al. 2012) and seeds were obtained by growing the parental generation of all lines used side by side in the same growth chambers under the same conditions ((Petrascheck et al., 2013). To standardize the input of microbes from the soil, seeds were propagated for three generations on the same commercial soil (Sun Gro, Agraham, USA). Moreover, to avoid the transmission of epiphytic microbes from soil, each generation of seeds was surface sterilized in 2.8% hypochlorite for 4 min and washed three times in MonoQ sterile water before seeding.

2. Proposed model in Figure 7

As the study does not investigate whether the seed microbiome influences plant fitness or health, the proposed model should more strictly reflect the findings presented in this manuscript. I suggest focusing the model on host genetic regulation of microbiome composition rather than broader implications.

Thank you for the comment. We think that the model should capture both aspects: The potential role of RBP47 in the adaptation of the host to the specific environmental conditions of iron toxicity AND the role that RBP47 plays in shaping the microbiome. We agree that this is a challenging model but should provide a good basis for further research into the single or dual role of RBP47 in the adaptation of Arabidopsis to the soil and geo-climatic conditions.

(Minor comments)

- The resolution of the supplemental figures is too low to interpret. Please provide higher-resolution versions.

Corrected.

- L215: Should "Fig. S2" be "Fig. 1"?

Thank you for your comment. Fig EV2(A) shows the relative abundance of the top 10 genera across all Arabidopsis accessions.

- L220-221: The sentence refers to "large sample size" and "fewer samples" in relation to diversity, but the logic is unclear. Please clarify how these factors affect microbial diversity.

We thank the reviewer for this observation. Our intention was not to suggest a causal relationship between sample size and microbial diversity, but rather to highlight an interesting

contrast between countries with different sampling depths. Despite Sweden having a larger number of accessions, its average Shannon diversity remained moderate, whereas France, with fewer accessions, displayed greater variability in diversity. We have revised the text to clarify this point and avoid misinterpretation.

- L338: Please remove the redundant word "most" ("the most most abundant").

Corrected.

Referee #3:

This study investigated the diversity of Arabidopsis seed microbiomes derived from multiple countries. They identified some key bacteria taxa like Delftia, Stenotrophomonas and Chryseobacterium in certain Arabidopsis accessions were also correlated with the whole seed microbiomes diversity. Two predicted functional KO terms shared by these key bacteria taxa were identified which are related to the iron metabolism of the bacteria. Then the authors' GWAS study identified RBP47B as key genes regulating the host-bacteria interaction in seeds, which is really cool in plant microbiome field. The seed microbiome between Col-0 and rbp47abcc' quadruple mutants indeed exist very drastic changes, which confirms RBP47B can shape the seed microbiome of Arabidopsis. The phenotypic assays using Col-0 and rbp47abcc' quadruple mutants under low pH and high iron concentration also demonstrated that RBP47B is crucial in helping Arabidopsis to adapt high iron and low pH condition. This study unveil a novel insight into how seed microbiome are assembled in different accessions, and the underlying critical genetic mechanisms.

Overall this is a great and successful population level GWAS study, and would also be extremely valuable for the community to further explore genetic regulations for root microbiome. Even though I have two major concerns related to the project:

We thank the reviewer for the overall assessment, and we appreciate the time and effort taken to evaluate our manuscript. We respectfully acknowledge and hereby respond the concerns raised to improve the clarity of the study.

A. It is not clear that why correlations between taxon abundance and beta diversity can be used to determine "key taxa regulate seed microbiome diversity"? It seems to be purely a mathematic relationship. Are there any microbial ecological theory or tools to dissect "key taxa regulating diversity"?

Thank you for the comment. Our goal in identifying these taxa was not to claim direct regulation in the manuscript, but to highlight taxa their presence or abundance is consistently associated with shifts in community structure across accessions, suggesting that they may play influential or indicator roles within the seed microbiome.

In line 312, what does the authors mean "core taxa"? Overall it seem vague here how to define "key taxa (Line 285)" or "core taxa (Line 312)"? I would suggest to consider the co-occurrence networks for each ecotype, and set a exact degree cutoff or cutoff for other other

microbial network topology indexes? That would be more accurate and non-biased for the define of any "core" or "key" stuff.

We apologize for confusing terms. We now consistently use the term “key taxa”.

We also taken a approach of network analysis in our manuscript. By calculating the correlation (Spearman) between the abundance of each taxon having a significant correlation with beta diversity we build a correlation network and identified this taxa with high degree of connectivity (Dataset EV3 and Fig EV4) so if a taxon’s abundance strongly influences beta diversity, it likely modulates interactions within the community and is thereby called “key taxa”.

B.The authors used PICRUSt based strain function prediction as a input for GWAS study, this could be problematic. Firstly, the golden standard for microbiome functional gene analyses is to conduct metagenome study. For a large amount of ecotype samples, it is also fine to predict microbiome functions for descriptive discussion, however, it would be challenging to use predicted microbiome functions for accurate GWAS inputs? Secondly, diverse GWAS or genetic mutant studies suggested that plant genes can regulate certain microbiome taxa abundance. I am wondering why the authors did not use the taxa abundance data as a GWAS input? This seems more straitforward to use abundance as GWAS input.

Yes, your understanding is absolutely correct. We first performed a GWAS using the relative abundance of the *trio* as the phenotype. Then, we explored the common KEGG Orthology (KO). Finally, we conducted a second GWAS using the predicted KO functional profiles (weighted normalized value) as phenotypes.And then we only show a SNP which was common to triad abundance and KO.(SNP 6617337_3 and 6614038_3) and tested its experimental validation.We have now included the GWAS output with other genera too in Dataset EV4.

Minor points:
1.Separation of microbiome samples according to different countries could be interesting, however, is not of any biological and ecological implications (country bounders does not necessary mean ecological differences). The author could avoid too much discussion based on different countries.

Our explanation here is consistent with the clarification provided in response to Reviewer 2.

2.Keywords: low pH and iron toxicity could be replaced with other words more related to the key finding of the manuscript, like stress fitness, or ribosomal proteins

Thank you, we revised the text.

3.Line 48, grammar mistake (with.....), Line48, 51,53, need references.

Thank you, we revised the text.

4.Line 54, "alter microbial community" is not equal to "dysbiosis".

Thank you, we revised the text.

5.Line 243, wrong legend description about color coding.

Thank you, we revised the text.

6.Line 336, two "most" repeated

Thank you, we revised the text.

7.Line 336 said "most abundant ASVs", but fig 4a seems to be KO functions rather than ASVs? This is very important result and should be checked.

8.line 406, "rbp47abcc" should be "RBP47B", or "rbp47abcc mutant affects"

Thank you, we revised the text.

9. Line 417-419, did the authors provided exact data and the analysis method for this critical correlation results?

The data were retrieved from the European Soil Database ESDAC and can be obtained at https://esdac.jrc.ec.europa.eu/Library/Data/PH/Documents/pH_Pub.pdf for pH and https://esdac.jrc.ec.europa.eu/ESDB_Archive/eusoils_docs/other/EUR23439.pdf; <https://geoera.eu/themes/raw-materials/> for iron.

10. The identification of RBP47B protein as a regulator of seed microbiome is really cool. The function of ribosomal proteins in root-microbe interactions was also revealed by a recent study [PMID: 40175371], and could be discussed to better support the robustness of this GWAS finding and genetic validation. Meanwhile, it would be interesting to further study whether rbp47 multiple mutants can block certain microbes mediated iron or pH stress damage.

It is interesting to note that. We have revised the discussion as suggested. In addition, we indeed attempted to study the beneficial effect of microbes (triad) isolated from the wild-type to rescue the phenotype in *rbp47abcc*' but we could not isolate the triad microbes in monoculture.

11.In agriculture, the most detrimental effect of low pH on crops is usually the secondary aluminium stress. Does RBP47B broadly affect root tolerance to both iron stress and aluminium stress? Whether RBP47B regulated microbiome confers fitness benefits for plants under iron or aluminium stresses

Thank you for indicating aluminium stress as well. We have not considered aluminium but will do so in future investigations.

12.Line 393, which taxa?

Triad1 (Delftia, Chrysobacterium and Stenotrophomanus) has been included in the text.

References:

- Anderson, M. J., Crist, T. O., Chase, J. M., Vellend, M., *et al.* (2011). Navigating the multiple meanings of β diversity: A roadmap for the practicing ecologist. *Ecology Letters*, *14*(1), 19–28.
- Brachi, B., Filiault, D., Whitehurst, H., Darme, P., Le Gars, P., Le Mentec, M., Morton, T. C., Kerdaffrec, E., Rabanal, F., Anastasio, A., Box, M. S., Duncan, S., Huang, F., Leff, R., Novikova, P., Perisin, M., Tsuchimatsu, T., Woolley, R., Dean, C., Nordborg, M., Holm, S., & Bergelson, J. (2022). Plant genetic effects on microbial hubs impact host fitness in repeated field trials. *Proceedings of the National Academy of Sciences of the United States of America*, *119*(30), e2201285119. <https://doi.org/10.1073/pnas.2201285119>
- Douglas, G. M., Maffei, V. J., Zaneveld, J. R., Yurgel, S. N., Brown, J. R., Taylor, C. M., Huttenhower, C., & Langille, M. G. I. (2020). PICRUSt2 for prediction of metagenome functions. *Nature Biotechnology*, *38*, 685–688. <https://doi.org/10.1038/s41587-020-0548-6>
- Horton, M. W., Hancock, A. M., Huang, Y. S., Toomajian, C., Atwell, S., Auton, A., Muliyati, N. W., Platt, A., Sperone, F. G., Vilhjálmsson, B. J., Nordborg, M., Borevitz, J. O., & Bergelson, J. (2012). Genome-wide patterns of genetic variation in worldwide *Arabidopsis thaliana* accessions from the RegMap panel. *Nature Genetics*, *44*(2), 212–216. <https://doi.org/10.1038/ng.104>
- Kreft, H., & Jetz, W. (2007). Global patterns and determinants of vascular plant diversity. *Proceedings of the National Academy of Sciences of the United States of America*, *104*(14), 5925–5930. <https://doi.org/10.1073/pnas.0608361104>
- Langille, M. G. I., Zaneveld, J., Caporaso, J. G., McDonald, D., Knights, D., Reyes, J. A., Clemente, J. C., Burkepile, D. E., Vega Thurber, R. L., Knight, R., Beiko, R. G., & Huttenhower, C. (2013). Predictive functional profiling of microbial communities using 16S rRNA marker gene sequences. *Nature Biotechnology*, *31*(9), 814–821. <https://doi.org/10.1038/nbt.2676>
- Legendre, P., & Anderson, M. J. (1999). Distance-based redundancy analysis: Testing multispecies responses in multifactorial ecological experiments. *Ecological Monographs*, *69*(1), 1–24. [https://doi.org/10.1890/0012-9615\(1999\)069\[0001:DBRATM\]2.0.CO;2](https://doi.org/10.1890/0012-9615(1999)069[0001:DBRATM]2.0.CO;2)
- Legendre, P., & Legendre, L. (2012). *Numerical ecology* (3rd English ed.). Elsevier.

- Lozupone, C., & Knight, R. (2005). UniFrac: A new phylogenetic method for comparing microbial communities. *Applied and Environmental Microbiology*, 71(12), 8228–8235.
- Meijón, M., Satbhai, S., Tsuchimatsu, T., *et al.* (2014). Genome-wide association study using cellular traits identifies a new regulator of root development in *Arabidopsis*. *Nature Genetics*, 46, 77–81. <https://doi.org/10.1038/ng.2824>
- Orme, C. D. L., Davies, R. G., Burgess, M., Eigenbrod, F., Pickup, N., Olson, V. A., Webster, A. J., Ding, T.-S., Rasmussen, P. C., Ridgely, R. S., Stattersfield, A. J., Bennett, P. M., Blackburn, T. M., Gaston, K. J., & Owens, I. P. F. (2005). Global hotspots of species richness are not congruent with endemism or threat. *Nature*, 436, 1016–1019. <https://doi.org/10.1038/nature03850>
- Pandey, P., Tripathi, A., Dwivedi, S., Lal, K., & Jhang, T. (2023). Deciphering the mechanisms, hormonal signaling, and potential applications of endophytic microbes to mediate stress tolerance in medicinal plants. *Frontiers in Plant Science*, 14, 1250020. <https://doi.org/10.3389/fpls.2023.1250020>
- Pascale, A., Proietti, S., Pantelides, I. S., & Stringlis, I. A. (2020). Modulation of the root microbiome by plant molecules: The basis for targeted disease suppression and plant growth promotion. *Frontiers in Plant Science*, 10, 1741. <https://doi.org/10.3389/fpls.2019.01741>
- Pereira, H. M., Ferrier, S., Walters, M., Geller, G. N., Jongman, R. H. G., Scholes, R. J., Bruford, M. W., Brummitt, N., Butchart, S. H. M., Cardoso, A. C., Coops, N. C., Dulloo, E., Faith, D. P., Freyhof, J., Gregory, R. D., Heip, C., Höft, R., Hurtt, G., Jetz, W., ... Wegmann, M. (2013). Essential biodiversity variables. *Science*, 339(6117), 277–278. <https://doi.org/10.1126/science.1229931>
- Ramette, A. (2007). Multivariate analyses in microbial ecology. *FEMS Microbiology Ecology*, 62(2), 142–160.
- Rodrigues, A. S. L., Pilgrim, J. D., Lamoreux, J. F., Hoffmann, M., & Brooks, T. M. (2006). The value of the IUCN Red List for conservation. *Trends in Ecology & Evolution*, 21(2), 71–76. <https://doi.org/10.1016/j.tree.2005.10.010>

Dear Prof. Hirt

Thank you for the submission of your revised manuscript to our offices. We have now received the enclosed reports from two of the original referees that were asked to assess it. EMBOR-2025-62320V2 still has minor suggestions that I would like you to address and incorporate before we can proceed with the official acceptance of your manuscript. Additionally, I provide below technical and formatting comments from our editorial assistance team that must be addressed before formal acceptance.

I look forward to seeing a new revised version of your manuscript as soon as possible.

Yehu Moran
Academic Editor
EMBO Reports

comments by editorial assistance team
(Bojana Perkucin 10th Sep 25) ---

BP 4.9.25

MANUSCRIPT FORMAT: Error - main, EV and suppl. figures have been provided in the manuscript; all figures should be removed and only main and EV figure legends should remain and the end of the manuscript.

Data Availability Statement: in, but needs to be renamed to Data Availability

Conflict of Interest: in, but it needs to be renamed to Disclosure and Competing Interests Statement

AUTHORS: OK

Author Contributions/CRedit: need to be removed from the manuscript text and provided only via the submission system.

FUNDING INFO: OK, but the separate section is not needed, this info should be only listed under Acknowledgments

FIGURES IN SEPARATE FILES: There are 7 main figures in the manuscript and only 5 have been uploaded separately; Figure files of EV figures should not have any legends/text embedded. Please correct.

APPENDIX FILE WITH ToC: not provided but there are 2 Appendix figures uploaded separately; these should either be provided in a PDF titled Appendix (with a title page and ToC with page numbers) or as EV figures.

SYNOPSIS IMAGE: missing, please provide.

SYNOPSIS TEXT: missing, please provide.

R&T TABLE: missing, please provide.

SOURCE DATA: missing, requested 14th Aug 25 and then provided by author with completed checklist, but 6B folder is empty. Please check and correct.

- Materials & Methods should be renamed Methods

- The manuscript sections should be in the following order: Title page - Abstract & Keywords - Introduction - Results - Discussion - Methods - Data Availability - Acknowledgments - Disclosure Statement & Competing Interests - References - Figure Legends - (Main Tables with legends if applicable) - Expanded View Figure Legends.

- Ethics approval and consent to participate should either be removed from the manuscript or placed somewhere in the Methods section.

- Consent for publication should be removed from the manuscript.

Figure Legends - Comments

- Please note that the exact p values are not provided in the legend of figure 6C. Please provide.

- Please note that information related to n is missing in the legends of figures 6C. Please provide.

- Please note that the measure of center for the error bars needs to be defined in the legends of figures 6C. Please provide.

specific comments by referees

Referee #1:

Most of our comments were addressed. We identified one remaining issue:

The top 10 genera in Figure 2a are different from those in the first submission, for example, Pelagibacterales is now labeled as "unclassified." In Figure 2b, the relative abundances are entirely different from the first version. The description in the Results (lines 285-286) is also inconsistent with the figures. This raises some concern about the underlying data, please recheck the data and regenerate the plots.

Minor comment:

In the Keywords, there are two commas before "GWAS."

Referee #2:

The authors have adequately addressed my previous comments.

Point-by-point response to editorial and reviewer requests:

MANUSCRIPT FORMAT: Error - main, EV and suppl. figures have been provided in the manuscript; all figures should be removed and only main and EV figure legends should remain and the end of the manuscript : Done

Data Availability Statement: in, but needs to be renamed to Data Availability: Done

Conflict of Interest: in, but it needs to be renamed to Disclosure and Competing Interests Statement: Done

AUTHORS: OK

Author Contributions/CRedit: need to be removed from the manuscript text and provided only via the submission system. Done

FUNDING INFO: OK, but the separate section is not needed, this info should be only listed under Acknowledgments. Done

FIGURES IN SEPARATE FILES: There are 7 main figures in the manuscript and only 5 have been uploaded separately; Done

Figure files of EV figures should not have any legends/text embedded. Please correct. Done

APPENDIX FILE WITH ToC: not provided but there are 2 Appendix figures uploaded separately; these should either be provided in a PDF titled Appendix (with a title page and ToC with page numbers) or as EV figures. Done

SYNOPSIS IMAGE: missing, please provide. Done

SYNOPSIS TEXT: missing, please provide. Done

R&T TABLE: missing, please provide. Done

SOURCE DATA: missing, requested 14th Aug 25 and then provided by author with completed checklist, but 6B folder is empty. Please check and correct. Added

- Materials & Methods should be renamed Methods: Done

- The manuscript sections should be in the following order: Title page - Abstract & Keywords - Introduction - Results - Discussion - Methods - Data Availability - Acknowledgments - Disclosure Statement & Competing Interests - References - Figure Legends - (Main Tables with legends if applicable) - Expanded View Figure Legends. Done

- Ethics approval and consent to participate should either be removed from the manuscript or placed somewhere in the Methods section. Done

- Consent for publication should be removed from the manuscript. Done

Figure Legends - Comments

- Please note that the exact p values are not provided in the legend of figure 6C. Please provide.

- Please note that information related to n is missing in the legends of figures 6C. Please provide.

- Please note that the measure of center for the error bars needs to be defined in the legends of figures 6C. Please provide. Added

specific comments by referees

Referee #1:

Most of our comments were addressed. We identified one remaining issue:

The top 10 genera in Figure 2a are different from those in the first submission, for example, Pelagibacterales is now labeled as "unclassified." In Figure 2b, the relative abundances are entirely different from the first version. The description in the Results (lines 285-286) is also inconsistent with the figures. This raises some concern about the underlying data, please recheck the data and regenerate the plots.

We appreciate the reviewer's observation regarding the mismatch between the "top 10" genera reported per accession versus after aggregation by country. In the initial submission, the ranking inadvertently mixed criteria (relative abundance in some views versus frequency of occurrence in others), which could yield apparent discrepancies across stratifications. In the revision, we redefined the "top 10" using a single, consistent rule based on prevalence across accessions (actual occurrence), and then reported the relative abundance of this fixed set of genera at both the accession and country levels. With this harmonized approach, the distributions are now internally consistent across both stratifications. Importantly, these changes do not affect our main conclusions; they improve clarity and reproducibility.

Minor comment:

In the Keywords, there are two commas before "GWAS." Done

Referee #2:

The authors have adequately addressed my previous comments.

Prof. Heribert Hirt
King Abdullah University of Science and Technology
Center for Desert Agriculture
4700 King Abdullah University of Science and Technology
Thuwal
Saudi Arabia

Dear Prof. Hirt,

I am very pleased to accept your manuscript for publication in the next available issue of EMBO reports. Thank you for your contribution to our journal.

Yours sincerely,

Yehu Moran
Editor
EMBO Reports
